

# A constructive theory of the numerically accessible many-body localized to thermal crossover

**Philip J D Crowley[1*] and Anushya Chandran[2]**

**1** Department of Physics, Massachusetts Institute of Technology,
Cambridge, Massachusetts 02139, USA
**2** Department of Physics, Boston University, Boston, MA 02215, USA

⋆ philip.jd.crowley@gmail.com

## Abstract

The many-body localised (MBL) to thermal crossover observed in exact diagonalisation studies remains poorly understood as the accessible system sizes are too small to be in an asymptotic scaling regime. We develop a model of the crossover in short 1D chains in which the MBL phase is destabilised by the formation of many-body resonances. The model reproduces several properties of the numerically observed crossover, including an apparent correlation length exponent $\nu = 1$, exponential growth of the Thouless time with disorder strength, linear drift of the critical disorder strength with system size, scale-free resonances, apparent $1/\omega$ dependence of disorder-averaged spectral functions, and sub-thermal entanglement entropy of small subsystems. In the crossover, resonances induced by a local perturbation are rare at numerically accessible system sizes $L$ which are smaller than a *resonance length* $\lambda$. For $L \gg \sqrt{\lambda}$ (in lattice units), resonances typically overlap, and this model does not describe the asymptotic transition. The model further reproduces controversial numerical observations which Refs. [1, 2] claimed to be inconsistent with MBL. We thus argue that the numerics to date is consistent with a MBL phase in the thermodynamic limit.



## Contents



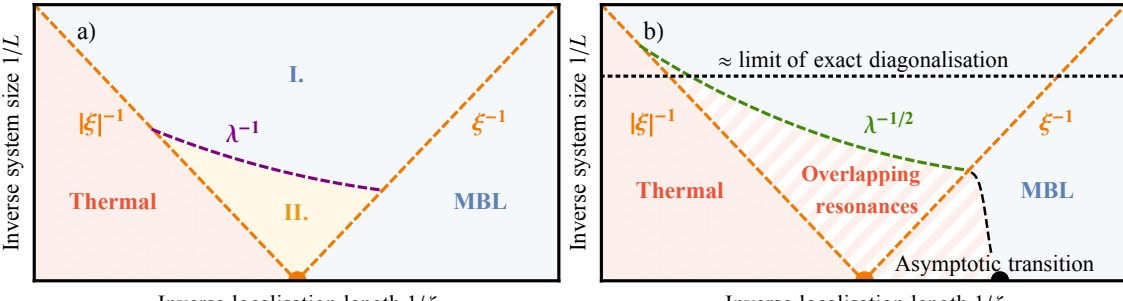

Figure 1: *a) The resonance model (RM)* predicts a continuous transition (orange point) between a localised (blue) and a thermal (red) phase, and an inverse correlation length $|\xi|^{-1}$ (orange lines) that vanishes with exponent $\nu = 1$ at the transition. At finite size, the transition is smeared into a crossover with 'fuzzy' boundaries (demarcated by the dashed orange lines). Within the crossover region the behaviour depends on whether the resonance length $\lambda$ (purple) is greater/lesser than the systems size $L$. In region I $L < \lambda$, and typical eigenstates have no resonances and spectrally averaged properties resemble those of the localised phase. *b) The MBL-thermal finite-size crossover*: At large $L$ in the vicinity of the RM transition (hatched region), localisation is inconsistent due to overlapping resonances. The RM is however self-consistent in the blue regions. The RM thus describes the MBL-thermal crossover in small system numerics (horizontal line), even though it does not describe the asymptotic transition (black point).

# 1 Introduction

Interacting one-dimensional quantum systems generically many-body localise (MBL) in the presence of strong disorder. Local subsystems of a MBL system do not thermalise; they instead retain memory of their initial conditions indefinitely. MBL thus provides a remarkable counterexample to the ergodic hypothesis, the cornerstone of quantum statistical mechanics [3–8], and allows for exotic quantum orders at finite energy densities [9–14, 14–17, 17–28].

Statistical descriptions of both the thermal and MBL phases have been corroborated by numerical studies. Specifically, the thermal phase is found to obey the eigenstate thermalisation hypothesis (ETH) [29–37], whereas the MBL phase violates the ETH and is instead characterised by a complete set of quasi-local conserved quantities (or l-bits) [38–44].

However, theoretical descriptions and numerical observations of the MBL-thermal transition remain at odds with one another. Phenomenological models suggest that the transition has Kosterlitz-Thouless-type scaling [45–47], and occurs when the localised phase is destabilised by rare thermal regions which seed "thermalisation avalanches" [48–54]. Numerical studies, which are limited to small systems, do not find any evidence of rare thermal regions [55, 56], but are known to be plagued by unexplained finite-size effects [57–60]. The absence of a theory of the finite-size crossover leaves unclear which features of the numerical data may survive in the thermodynamic limit, and has led Refs. [1, 2] to argue that finite size numerics is inconsistent with the existence of MBL in the thermodynamic limit.

We develop a microscopically motivated *resonance model* (RM) for the one-dimensional MBL-thermal crossover at finite sizes. In this model the MBL phase is not destabilised by rare thermal regions, but by many-body resonances involving macroscopically distinct l-bit states. Although this mode of instability was previously identified [61] and observed in finite size numerics [62], it has received little attention in the literature.

Specifically, we consider a presumptively many-body localised chain, analyse the statistics

of resonances induced by local perturbations, and establish when these resonances destabilise MBL. The detailed analysis is different in the Floquet (Sec. 2) and Hamiltonian (Sec. 3) settings. However, in both cases, the same set of non-trivial length scales emerge which control the physics. The first of these is the bare localisation length $\zeta$, which governs the exponential decay of off-diagonal matrix elements of local operators in the l-bit basis. A site-local perturbation introduces many-body resonances between eigenstates. The probability that a given eigenstate finds a first-order resonance involving l-bits within a range $r$ (in the Floquet case) is given by

$$q(r) = \frac{e^{-r/\xi}}{\lambda} \, . \tag{1}$$

Here, two additional lengths emerge: the *correlation length* $\xi$ sets the typical range of resonances, while the *resonance length* $\lambda$ determines their density. The RM predicts that $\xi$ diverges as the localisation length approaches the critical value $\zeta_c$. This marks the transition between a localised phase in which the number of resonances is finite and a delocalised phase (dubbed thermal in Fig. 1a) in which the number of resonances grows exponentially with range. The finite-size behaviour near the transition depends crucially on the resonance length $\lambda$ which is much larger than the lattice scale. For system size $L \ll \lambda$ (region I, Fig. 1a), typical eigenstates have no resonances and non-thermal expectation values. For system sizes $L \gg \lambda$ (region II), typical states participate in $L/\lambda \gg 1$ resonances even at first-order [1].

The first-order analysis is clearly incomplete in regimes where the number of resonances induced by a single local perturbation grows with $L$ (region II and thermal). In fact, the region of instability is somewhat larger if we consider locally perturbing the system at every site. In this case, a typical eigenstate develops a density $\sim \xi/\lambda$ of resonances each of which rearranges a region of size $\xi$ (here and henceforth we measure lengths in units of the lattice constant). For $\xi \gtrsim \sqrt{\lambda}$, the resonances typically spatially overlap and we expect them to lead to l-bit rearrangements on the scale of the system. The hatched region in Fig. 1b indicates the parameter regime and finite sizes where localisation in the RM is inconsistent due to this instability.

Nevertheless, we present analytical arguments in Sec. 4 that the RM is self-consistent outside of the hatched region – i.e. at small enough $L$ in region I and at any $L$ for large enough disorder (i.e. $1/\zeta$). Rough estimates of the resonance length in Floquet and Hamiltonian disordered chains suggest $15 \lesssim \lambda \lesssim 50$ for models numerically studied to date (see Sec. 4) . Thus, we believe that numerically accessible system sizes correspond to the horizontal dashed line in Fig. 1b, so that the observed crossovers in spectral quantities, spectral functions, finite-size drifts, etc. can all be predicted within the region of validity of the RM. Summarising the more detailed results in Sec. 5, the RM reproduces many features of numerically exact data:

- *Localised region I:* As typical eigenstates do not find a resonance for $L \ll \sqrt{\lambda}$, the RM predicts that region I displays the phenomenology of the localised phase: long-time local memory, a logarithmically growing light cone, sub-thermal eigenstate entanglement entropy of small sub-systems etc.. Spectrally averaged quantities are thus insensitive to the boundary between the MBL phase and region I ($\xi = L$, Fig. 1b), in agreement with Ref. [58].

- *Correlation length exponent $\nu$:* The correlation length exponent in the RM is given by $\nu = 1$, consistent with the values extracted from finite-size scaling in ED [8, 11, 59, 63]. Note that $\nu = 1$ violates the Harris criterion [57, 64, 65].

---

[1] Naively, region II is the 'critical fan' in which $\xi \gg L \gg$ all other length scales. However, we refrain from this nomenclature as the region is masked by the collective instability of overlapping resonances discussed next.

- *Drift of the critical disorder strength $W_c$ with L:* The RM predicts the controversial observation of Refs. [1, 2] that $W_c \propto L$ at small $L$.

- *Apparent $1/\omega$ low-frequency dependence of spectral functions:* In region I, disorder-averaged spectral functions $[S(\omega)]$ exhibit a low-frequency power-law divergence with a continuously varying exponent (throughout we use $[\cdot]$ to denote disorder averaging). The divergence is strongest in the middle of region I, with $[S(\omega)] \sim 1/\omega^{1-\theta_c}$ (Floquet, Fig 2a–b) or $[S(\omega)] \sim 1/\omega|\log\omega|^{1/2}$ (Hamiltonian, Fig 2d–e). As the corrections are small ($\theta_c \ll 1$), the RM explains the apparent $1/\omega$ behaviour reported in Refs. [2, 66].

- *Scale-free resonances:* Within regions I and II, $q(r)$ is scale-invariant and resonances form at all ranges, in agreement with a numerically exact calculation of $q(r)$ [62].

- *Apparent sub-diffusion:* On the thermal side of the transition ($0 < -\xi < L$), the dynamics at short times $t < \omega_\xi^{-1}$ is critical. The RM describes this dynamics, and predicts a continuously varying exponent $z$ in spectral functions $\sim 1/\omega^{1-1/z}$ (see Figs 2a–b for Floquet, and Figs 2d–e for the Hamiltonian case). The RM thus explains the apparent sub-diffusion (as measured by $z$) reported in several studies [2, 66–70], without invoking rare region effects, which Ref. [55] finds are absent in numerically accessible systems.

- *Exponential increase of Thouless time at weak disorder $W \ll W_c$:* This numerical observation of Refs. [1, 2] follows from the logarithmic growth of the light cone until time $t \approx \omega_\xi^{-1}$ in the thermal phase of the RM.

As the resonance model of the finite-size crossover assumes the existence of MBL, and reproduces the numerical observations of Refs [1, 2], we conclude to their contrary, that the numerics to date appears consistent with a stable MBL in the thermodynamic limit.

We additionally predict three interesting features of the dynamical phase diagram that could be tested numerically in the near future.

- The exponents controlling the strongest low-frequency divergence of $[S(\omega)] \sim 1/\omega^{1-\theta_c}$ in region I: We predict that the exponent $\theta_c$ is a non-zero *non-universal* value in the Floquet setting, while $\theta_c \to 0^+$ (corresponding to log corrections) in the Hamiltonian setting with energy conservation. That is, the existence and number of conservation laws affects the scaling theory of the finite-size MBL-thermal crossover.

- An empirical criterion for MBL: In localised systems, the distribution $\varrho(v)$ of matrix elements of a local operator $V$ that couple eigenstates in two small non-overlapping mid-spectrum energy (or quasi-energy) windows takes the form,

$$\varrho(v) \sim v^{-2+\theta_0}, \tag{2}$$

with $0 < \theta_0 \le 1$ (see Fig. 2c). A simple numerical criterion follows:

$$\rho\bar{v} \sim 2^{L/2} \text{ (thermal)}, \quad \rho\bar{v} \sim \text{cons. (MBL)}, \tag{3}$$

with $\rho$ denoting the mid-spectrum many-body density of states. This criterion generalises the avalanche stability criterion of Ref. [48] to a setting without l-bits or rare thermalising regions.

- Detecting the crossover between MBL and region I: In region I, scale free resonances form, but remain rare. Thus eigenstate averaged observables are largely insensitive to the formation of resonances. However, by analysing the distribution of an observable

over eigenstates, or conditioning on the formation of resonances, it is possible to numerically detect the crossover between MBL and region I. Such an analysis is performed in Ref. [62].

We proceed as follows. In Section 2, we describe the Floquet resonance model, couple the RM to a probe spin, compute the statistics of many-body resonances that a reference l-bit state is involved in, and thus derive the disorder-averaged spectral function of a local operator. In Sec. 3 we repeat the analysis for a Hamiltonian system. In Sec. 4 we establish the regime in which the RM is self-consistent, showing it to apply to small and strongly disordered systems (small $L$ in region I in Fig. 1). In Sec. 5 we discuss the implications of this analysis for interpreting finite-size numerical data, before concluding in Sec. 6.

## 2 Floquet resonance model

We now describe the computation of the spectral function $[S(\omega)]$ of a local operator within the RM for a Floquet-MBL system. Specifically we calculate $[S(\omega)]$ for a Pauli operator acting on an ancillary probe spin, coupled to an infinite MBL chain (Fig. 3). This simplifies the analysis as the the probe spin may be isolated without cutting the chain. The low frequency behaviour of $[S(\omega)]$ is though universal, and thus holds for any local operator defined on the chain. We first introduce the Floquet RM and define the localisation length $\zeta$ in Sec. 2.1, we then detail a careful counting of resonances induced by a probe spin in Sec. 2.2. Panels (a), (b) and (f) in Fig. 2 summarise the results for the spectral function of the probe spin in the Floquet RM.

Resonances do not span the system for $1/\zeta > 1/\zeta_c := \log 2$; this is the MBL phase of the Floquet RM. The RM MBL phase has infinite time memory of initial conditions, and a power-law divergence of the spectral function at small frequency (53).

The point $1/\zeta = 1/\zeta_c$ marks the transition out of the RM MBL phase, at which resonances occur on all length scales. The statistics of the strongest resonances determine the low-frequency scaling of $[S(\omega)]$ in regions I and II within Fig. 1a. The exponent $\theta$ characterising the low-frequency divergence of $[S(\omega)]$ in region II jumps at the transition (56).

Although typical states find increasingly many resonances at long ranges for $1/\zeta < 1/\zeta_c$, they remain rare on the scale of the correlation length $\xi$. Consequently, the RM predicts the behaviour of $[S(\omega)]$ at intermediate frequencies (58) in the thermal phase.

### 2.1 Set-up

#### 2.1.1 Chain Hamiltonian

Consider a generic strongly disordered and interacting quantum spin chain with periodic boundary conditions, and subject to a periodic (Floquet) drive. For example, the Heisenberg model with random $O(3)$ fields:

$$H(t) = \begin{cases} H_W = W \sum_n \boldsymbol{v}_n \cdot \boldsymbol{\sigma}_n, & 0 \leq \Omega t < \pi, \\ H_J = J \sum_n \boldsymbol{\sigma}_n \cdot \boldsymbol{\sigma}_{n+1}, & \pi \leq \Omega t < 2\pi, \end{cases} \tag{4}$$

where $W$, $J$ and $\Omega$ set the disorder strength, interaction strength and fundamental frequency of the drive respectively, $\boldsymbol{\sigma}_n = (\sigma_n^x, \sigma_n^y, \sigma_n^z)$ is the usual vector of Pauli matrices acting on the $n$th site, and $\boldsymbol{\sigma}_{L+1} = \boldsymbol{\sigma}_1$ enforces periodic boundary conditions. The $\boldsymbol{v}_n$ are independent and identically distributed (iid) random vectors with zero mean $[\boldsymbol{v}_n] = \boldsymbol{0}$ and unit variance

$[\boldsymbol{v}_n \cdot \boldsymbol{v}_n] = 1$, with, for example, iid Gaussian distributed entries. Here $[\cdot]$ denotes disorder averaging.

We assume two key properties of $H(t)$: (i) it has no global conservation laws, and (ii) for some finite $\Omega, W \gg J$, the model is Floquet many-body localised, as per Ref. [71]. The specific form of $H(t)$ is otherwise unimportant.

The dynamics of the chain is characterised by the Floquet operator

$$U_{\mathrm{F}} = \mathcal{T}\exp\left(-\mathrm{i}\int_0^T H(t)\mathrm{d}t\right) = \exp\left(-\mathrm{i}H_J T/2\right)\exp\left(-\mathrm{i}H_W T/2\right), \tag{5}$$

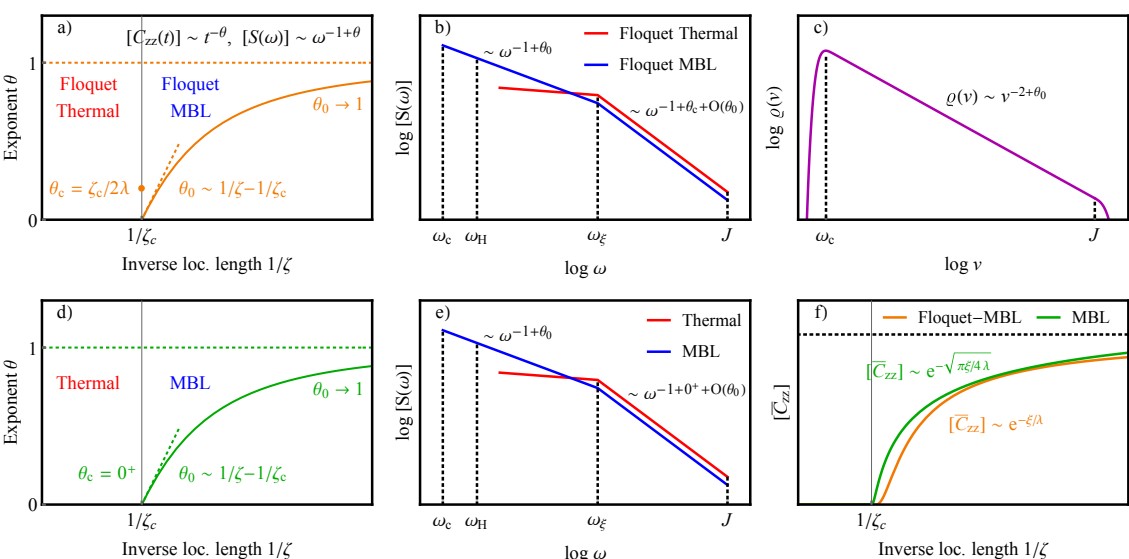

Figure 2: *Properties of the Resonance Model transition*: *Panels (a) and (d):* In the MBL phase and at the RM transition ($1/\zeta \geq 1/\zeta_{\mathrm{c}}$), the spectral function diverges at low frequencies $[S(\omega)] \sim \omega^{-1+\theta}$. Panels (a) and (d) summarise the behaviour of the exponent $\theta$ in the Floquet and Hamiltonian cases respectively. Both panels show $\theta = \theta_0 \to 1$ deep in the MBL phase ($1/\zeta \to \infty$), and $\theta \to 0$ as the transition is approached. At the Floquet RM transition, $\theta$ jumps to a finite value $\theta = \theta_{\mathrm{c}}$ (orange point, panel (a)), while $\theta_{\mathrm{c}} = 0^+$ (indicating the presence of log corrections) at the Hamiltonian RM transition. *Panels (b) and (e):* In the vicinity of the RM transition, the correlation length $|\xi|$ sets the cross-over frequency scale $\omega_\xi \sim \exp(-1/|\theta_0|)$. The low-frequency behaviour ($\omega \ll \omega_\xi$) is determined by the phase, while the intermediate frequency behaviour $\omega \gg \omega_\xi \gg J^{-1}$ is determined by the transition. The two other frequency scales are set by the system size: the Heisenberg scale $\omega_{\mathrm{H}}$ is the inverse level spacing, while $\omega_{\mathrm{c}}$ is the scale of the smallest off-diagonal matrix elements. The thermal-region I crossover occurs when $\omega_\xi \sim \omega_{\mathrm{H}} \sim \omega_{\mathrm{c}}$. In region I, only the exponent controlling the $\omega > \omega_\xi$ decay is visible. This exponent is continuously varying and is significantly corrected from its value at the transition in region I (as quantified by the $O(\theta_0)$ term). The smallest value of the exponent is however set by $\theta_{\mathrm{c}}$. *Panel (c):* The exponent $\theta_0$ may be directly extracted from $\varrho(\nu)$, the distribution of off-diagonal matrix elements of a local operator. In the localised phase, there are exponentially many off-diagonal matrix elements which are exponentially small in range, so $\varrho(\nu)$ diverges as a power-law at small $\nu$. The exponent defines $\theta_0$. *Panel (f):* The time averaged correlator $[\overline{C}_{zz}]$ serves as an order parameter for the MBL phase. $[\overline{C}_{zz}]$ goes to zero smoothly as $1/\zeta \to 1/\zeta_{\mathrm{c}}$ is approached from the MBL side, faster than any power law in both the Hamiltonian and Floquet cases.

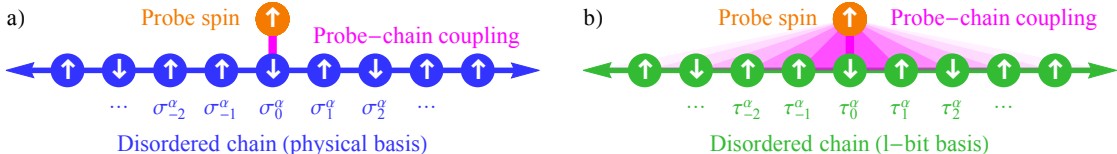

Figure 3: *Set-up in the physical and l-bit bases respectively*: a) A "probe" spin-$\frac{1}{2}$ (orange) couples to a strongly disordered chain (blue) at the site $n = 0$ (magenta). b) Transforming to the l-bit basis renders the Floquet unitary of the chain diagonal and the probe-chain coupling quasi-local. The coupling strength decays exponentially with distance from $n = 0$.

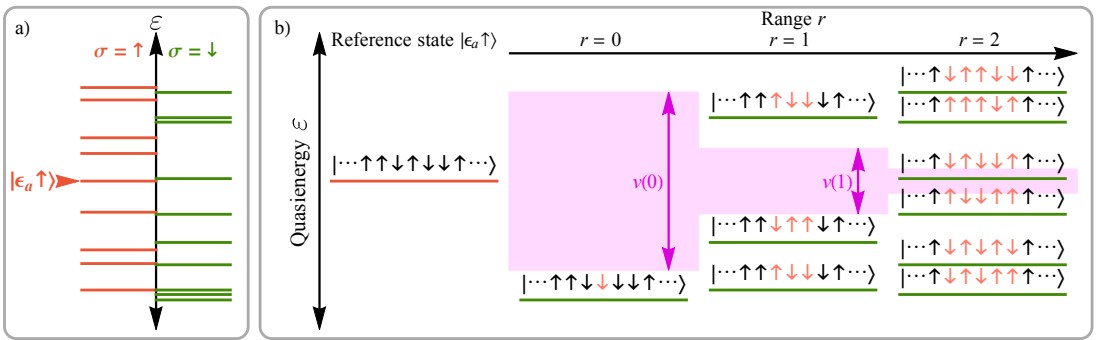

Figure 4: *Organising resonances by range*: a) The many-body spectrum of $\mathcal{H}_0$ in a small quasi-energy window is divided into two sectors labelled by the state of the probe spin $\sigma = \uparrow, \downarrow$. $|\epsilon_a \uparrow\rangle$ labels a specific reference state. b) The l-bit configuration corresponding to reference state (red spectral line) is shown. The states $|\epsilon_d \downarrow\rangle$ in the opposite sector (green lines) can be grouped according to their range $r$ from the reference state (ranges $r = 0, 1, 2$ shown); states at range $r$ differ only on the l-bits with index $|n| \leq r$ (highlighted in orange). A state $|\epsilon_d \downarrow\rangle$ at range $r$ is resonant with $|\epsilon_a \uparrow\rangle$ if its quasi-energy separation is less than the matrix element size $v(r)$ (i.e. if it lies within the magenta region). In the plot, the first resonance occurs at range $r = 2$.

where $T = 2\pi/\Omega$ and $\mathcal{T}$ is the usual time ordering operator. The associated Floquet states $|\epsilon_a\rangle$, and quasi-energies $\epsilon_a$ are defined by

$$U_{\mathrm{F}}|\epsilon_a\rangle = \mathrm{e}^{-\mathrm{i}\epsilon_a T}|\epsilon_a\rangle. \tag{6}$$

### 2.1.2 Localisation in the l-bit basis

At sufficiently strong disorder in the MBL phase, we assume that the Floquet states $|\epsilon_a\rangle$ may be identified with configurations of quasi-local integrals of motion, or *l-bits* [7,38,39] (in Sec. 4.1, we discuss how this assumption may be relaxed). Each l-bit $\tau_n^z$ is traceless $\mathrm{tr}\left(\tau_n^z\right) = 0$, squares to the identity $(\tau_n^z)^2 = \mathbb{1}$, is exponentially localised around the physical site $n$, and commutes with the Floquet operator

$$[U_{\mathrm{F}}, \tau_n^z] = 0. \tag{7}$$

Each Floquet state $|\epsilon_a\rangle$ can be identified with an l-bit configuration $\boldsymbol{\tau}_a \in \{-1, 1\}^L$. The scalar element $\tau_{an} = \pm 1$ of $\boldsymbol{\tau}_a$ specifies the state of the $n$th l-bit:

$$\tau_n^z|\epsilon_a\rangle = \tau_{an}|\epsilon_a\rangle. \tag{8}$$

A quasi-local operator $U$ diagonalises the Floquet unitary, and maps the physical spin operators to l-bits,

$$U\tau_n^\alpha U^\dagger = \sigma_n^\alpha. \tag{9}$$

Thus the $\sigma_n^\alpha$ are similarly exponentially localised operators in the l-bit basis.

Consider two eigenstates $|\epsilon_a\rangle$, $|\epsilon_b\rangle$. We say two states differ at range $r_{ab}$ if the furthest flipped l-bit is at distance $r_{ab}$ from the site $n = 0$.

$$r_{ab} := \max\{|n| : \tau_{an} \neq \tau_{bn}\}. \tag{10}$$

The range is depicted in Fig. 4b. If the matrix element $V_{ab} := \langle\epsilon_a|V|\epsilon_b\rangle$ of an operator $V$ is non-zero, then $V_{ab}$ is also said to have range $r_{ab}$.

The length scale on which a physical spin operator is localised in the l-bit basis defines the localisation length $\zeta$. Consider a local operator $V$ acting on the physical site of index $n = 0$. The operator $V$ can be decomposed into a sum of terms of increasing range

$$V = \sum_{r=0}^{L/2} V_r \tag{11}$$

where all the non-zero matrix elements of $V_r$ have range $r$. The asymptotic decay of the norm of $V_r$ defines $\zeta$:

$$\log|V_r| \sim -\frac{r}{\zeta}. \tag{12}$$

We use the re-scaled Frobenius norm

$$|V_r| := \sqrt{\frac{1}{2^L}\operatorname{tr}\left(V_r^2\right)}, \tag{13}$$

as it is simple to calculate analytically, and captures the typical expectation value of an arbitrary vector $|\langle\psi|V_r|\psi\rangle| \approx |V_r|$.

### 2.1.3 Coupling a probe spin to the disordered chain

To probe the dynamical phase of the disordered chain, we introduce a probe spin-$\frac{1}{2}$ $\boldsymbol{\sigma}_P$ subject to a $z$-field of strength $W$. The combined Hamiltonian of the probe spin and disordered chain,

$$\mathcal{H}(t) = \mathcal{H}_0(t) + \mathcal{H}_1(t), \tag{14}$$

is periodic with fundamental frequency $\Omega$. Here $\mathcal{H}_0$ encodes the part of the Hamiltonian in which the probe spin and disordered chain are decoupled

$$\mathcal{H}_0(t) = H(t) \otimes \mathbb{1} + \frac{h}{2}\mathbb{1} \otimes \sigma_P^z, \tag{15}$$

and $\mathcal{H}_1(t)$ encodes their coupling. Throughout we use cursive letters to denote properties of the combined Hilbert space of the disordered chain and the probe spin, and italic letters to denote properties of the reduced Hilbert spaces. The spin and chain are coupled an interaction $\mathcal{H}_1$, we choose

$$\mathcal{H}_1(t) = \sum_{n\in\mathbb{Z}} \delta(n - t/T) V \otimes \sigma_P^x. \tag{16}$$

Here $V$ is some local operator which acts only on the $n = 0$ site of the chain, and which is assumed to have norm $|V| = J$, e.g. $V = J\sigma_0^x$.

The Floquet operator of the combined system is given by

$$\mathcal{U}_F = \mathcal{U}_1 \mathcal{U}_0 \tag{17}$$

where $\mathcal{U}_0$ is the Floquet unitary for $\mathcal{H}_1 = 0$, and $\mathcal{U}_1$ encodes the interaction

$$\mathcal{U}_0 = U_{\mathrm{F}} \otimes \exp\left(-\tfrac{\mathrm{i}}{2} W T \sigma_{\mathrm{P}}^z\right) \tag{18}$$

$$\mathcal{U}_1 = \exp\left(-\mathrm{i} T V \otimes \sigma_{\mathrm{P}}^x\right). \tag{19}$$

Each eigenstate of the unperturbed Floquet unitary $\mathcal{U}_0|\varepsilon_\alpha^0\rangle = \mathrm{e}^{-\mathrm{i}\varepsilon_\alpha^0 T}|\varepsilon_\alpha^0\rangle$ is a tensor product of a quasi-energy state of the disordered chain $|\epsilon_a\rangle$ and a $z$-polarised state of the probe spin $|\sigma\rangle$,

$$
\begin{aligned}
|\varepsilon_\alpha^0\rangle &:= |\epsilon_a \sigma\rangle := |\epsilon_a\rangle \otimes |\sigma\rangle, \\
\varepsilon_\alpha^0 &= \epsilon_a + \tfrac{1}{2}\sigma W,
\end{aligned}
\tag{20}
$$

where $\alpha = (a, \sigma)$ is a composite label.

## 2.2  Spectral function of $\sigma_{\mathrm{P}}^z$ in the RM MBL phase $\zeta < \zeta_{\mathrm{c}}$

Our aim is to calculate the disorder averaged infinite temperature $zz$ spin correlator,

$$[C_{zz}(t)] = \frac{1}{\mathcal{D}} \left[ \mathrm{tr}\left(\sigma_{\mathrm{P}}^z(t)\sigma_{\mathrm{P}}^z(0)\right) \right], \tag{21}$$

in the RM. Here the normalization by $\mathcal{D}$, the Hilbert space dimension, ensures that $[C_{zz}(0)] = 1$. For simplicity, we restrict to stroboscopic observations at the drive period $t \in T\mathbb{N}$. The Heisenberg operator $\sigma_{\mathrm{P}}^z(t)$ at integer periods is given by

$$\sigma_{\mathrm{P}}^z(nT) = (\mathcal{U}_{\mathrm{F}}^\dagger)^n \sigma_{\mathrm{P}}^z \mathcal{U}_{\mathrm{F}}^n. \tag{22}$$

The spectral function $[S(\omega)]$ is obtained by Fourier transformation of (21),

$$[C_{zz}(t)] = \int_{-\infty}^{\infty} \mathrm{d}\omega\, \mathrm{e}^{-\mathrm{i}\omega t} [S(\omega)]. \tag{23}$$

The basic steps in the calculation are as follows. We resolve the trace in the correlator (21) over the eigenstates $|\epsilon_a \sigma\rangle$ of $\mathcal{H}_0$, and argue in Sec. 2.2.1 that each term is well approximated by either unity or a pure tone:

$$\langle \epsilon_a \sigma | \sigma_{\mathrm{P}}^z(t)\sigma_{\mathrm{P}}^z(0)|\epsilon_a \sigma\rangle = \begin{cases} 1 & \text{(no resonance)} \\ \cos\left(|V_{ab}|t\right) & \text{(resonance)} \end{cases} \tag{24}$$

Above, $|V_{ab}|$ is the largest matrix element that couples $|\epsilon_a \sigma\rangle$ to a resonant state $|\epsilon_b \bar{\sigma}\rangle$ where $\bar{\sigma}$ is the opposite $z$-spin projection as compared to $\sigma$. Taking the matrix elements at range $r$ to have a characteristic scale $v(r)$, we obtain

$$[C_{zz}(t)] = [\overline{C}_{zz}] + \int_0^{L/2} \mathrm{d}r\, p(r) \cos(v(r)t) \tag{25}$$

where $p(r)$ is the probability (upon varying the initial state, and disorder realisation) that the resonant process with the largest matrix element is at range $r$, and

$$[\overline{C}_{zz}] := \lim_{T \to \infty} \frac{1}{T} \int_0^T \mathrm{d}t\, [C_{zz}(t)] = 1 - \int_0^{L/2} \mathrm{d}r\, p(r) \tag{26}$$

is the probability of no resonances. As $p(r)$ and $v(r)$ are exponentially decaying in $r$, we find that the spectral function is a power law at low frequencies,

$$[S(\omega)] \propto \omega^{-1+\theta}. \tag{27}$$

$$|\varepsilon_{\alpha,\beta}\rangle \propto \left| \cdots \right\rangle \pm \left| \cdots \right\rangle$$

Figure 5: *Cartoon of approximate eigenstates*: For the purposes of calculating the spectral function, the resonant eigenstates may replaced with cat states. Here the resonance is of range $r = 2$, so that only l-bits with indices $n \in \{-2,-1,0,1,2\}$ (red box) are reconfigured.

The exponent $\theta$ approaches zero as $\zeta \to \zeta_c^-$ from the localised side, but jumps to a non-zero $\theta_c$ precisely at the critical point $\zeta = \zeta_c$. Ref. [61] gave a similar resonance counting argument for the low frequency properties of the spectral function in the localised phase.

At each stage of this analysis we approximate eigenstates as being either product states of the probe spin and l-bits, or cat states in which the probe spin is flipped. A more sophisticated treatment would account for intermediate situations in which a "partial resonance" forms [72]. However, such refinements do not change the low frequency properties of the spectral function, which are of interest here.

We now detail how these results are obtained. The final expressions for the spin-spin correlator are given in Secs. 2.2.4, 2.2.5.

### 2.2.1 Contribution of a resonance to the spectral function

Let us define a resonance. Consider a Floquet state $|\varepsilon_\alpha\rangle$ of combined system,

$$\mathcal{U}_{\mathrm{F}}|\varepsilon_\alpha\rangle = \mathrm{e}^{-\mathrm{i}\varepsilon_\alpha T}|\varepsilon_\alpha\rangle. \tag{28}$$

Expanding these Floquet states to leading order in $V$, we obtain

$$|\varepsilon_\alpha\rangle = |\epsilon_a \uparrow\rangle + \sum_b \frac{\mathrm{i}V_{ba}T}{\mathrm{e}^{\mathrm{i}(\epsilon_a - \epsilon_b + h)T} - 1}|\epsilon_b \downarrow\rangle + \dots. \tag{29}$$

where $\alpha = (a, \uparrow)$ [2]. We define the two states $|\epsilon_a \uparrow\rangle$ and $|\epsilon_b \downarrow\rangle$ to be resonant if the first-order correction is large, that is, if

$$g_{ba} := \max_{n \in \mathbb{Z}} \left| \frac{V_{ba}}{\epsilon_a - \epsilon_b + h + n\Omega} \right| > 1. \tag{30}$$

If $g_{ba} < 1$ for all $b$, then we approximate $|\varepsilon_\alpha\rangle$ by the unperturbed eigenstate $|\epsilon_a \uparrow\rangle$.

If $g_{ba} > 1$ for a single $b$, then degenerate perturbation theory yields 'cat' Floquet states

$$|\varepsilon_{\alpha,\beta}\rangle = \frac{1}{\sqrt{2}}\left(|\epsilon_a \uparrow\rangle \pm |\epsilon_b \downarrow\rangle\right) + \mathrm{O}(g_{ba}^{-1}), \tag{31}$$

to good approximation (see Fig. 5). The two cat states (31) are split in quasi-energy by the matrix element $|V_{ba}|$,

$$|\varepsilon_\alpha - \varepsilon_\beta| = |V_{ba}| + O(|V_{ba}|g_{ba}^{-2}). \tag{32}$$

Ignoring the sub-leading corrections, we thus obtain

$$\langle \epsilon_a \uparrow |\sigma_{\mathrm{P}}^z(t)\sigma_{\mathrm{P}}^z(0)|\epsilon_a \uparrow\rangle = \cos\left(|V_{ba}|t\right), \ t \in T\mathbb{N}. \tag{33}$$

---

[2]Eq. (29) recovers the standard first-order term in Hamiltonian perturbation theory in the high-frequency limit $T \to 0$

The corresponding contribution to the spectral function is two delta function peaks at $\omega = \pm |V_{ba}|$. The absence of weight at zero frequency is a consequence of the equal amplitudes in the RHS of (31). We argue in Appendix A that extending this calculation to include a small non-zero weight at $\omega = 0$ does not alter the low frequency behaviour of the disorder-averaged spectral function.

If $g_{ba} > 1$ for multiple indices $b$, the eigenstates do not have the simple form in (31). Nevertheless, we argue in Appendix A that the *strongest resonance*, corresponding to the largest matrix element, sets the frequency of oscillation if $\zeta < \zeta_c$. That is,

$$\langle \epsilon_a \uparrow | \sigma_{\mathrm{P}}^z(t) \sigma_{\mathrm{P}}^z(0) | \epsilon_a \uparrow \rangle = \cos(\omega_{a\uparrow} t), \quad \text{for} \quad t \in T\mathbb{N}, \tag{34}$$

with

$$\omega_{a\uparrow} = \max\left\{ |V_{ba}| : g_{ba} > 1 \right\}. \tag{35}$$

In other words, for an initial state $|\epsilon_a \uparrow\rangle$, the probe spin oscillates at a frequency $\omega_{a\uparrow}$ for a window of time $t \gg \omega_{a\uparrow}^{-1}$, and thus the Fourier transform of (34) is sharply peaked at $\pm\omega_{a\uparrow}$. Analogous expressions for an initial state in the down sector are easily similarly.

### 2.2.2 The probability $q(r)$ of resonance at range $r$

We take all the matrix elements at range $r$ to have a single characteristic value $v(r)$ which is a monotonically decreasing function of the range $r$. This recasts the problem of finding the resonance with the largest matrix element as the problem of finding the resonance with the smallest range $r$. We now calculate $v(r)$, and subsequently the probability $q(r)$ of finding a resonance at range $r$.

As described in Sec. 2.1, $V = \sum_{r=0}^{L/2} V_r$ may be decomposed into terms of increasing range $r$ in the MBL phase. $V_r$ couples a given state $|\epsilon_a\rangle$ to $N_r$ other states $|\epsilon_b\rangle$ at range $r$, where

$$N_0 = 1, \qquad N_{r>0} = \tfrac{3}{2} \cdot 4^r. \tag{36}$$

The characteristic scale $v(r)$ of each matrix element is determined by,

$$|V_r|^2 = \frac{1}{\mathcal{D}} \sum_a \langle \epsilon_a | V_r^2 | \epsilon_a \rangle = N_r \cdot v(r)^2. \tag{37}$$

Using $|V_r| \sim J e^{-r/\zeta}$ we obtain

$$v(r) = \frac{|V_r|}{\sqrt{N_r}} \approx J e^{-r/\xi - 2r/\zeta_c}, \tag{38}$$

where the *correlation length* $\xi$ is defined by

$$\frac{1}{\xi} = \frac{1}{\zeta} - \frac{1}{\zeta_c}, \qquad \frac{1}{\zeta_c} = \log 2. \tag{39}$$

The omission of the unimportant pre-factor of $\sqrt{3/2}$ makes (38) approximate.

Two properties of $\xi$ are noteworthy. First, $\xi$ has the interpretation of a length only in the MBL phase of the RM, in which it is positive. Second, $\xi$ diverges as $\zeta \to \zeta_c^-$. When we use results of the RM to discuss the short-time dynamics as $\zeta \to \zeta_c^+$, we will be careful to use the absolute value of $\xi$.

Let $\rho(r)\mathrm{d}r$ denote the density of states per unit quasi-energy with range in the interval $[r, r + \mathrm{d}r]$; from here on we will coarse grain and treat the range $r$ as a continuous variable.

As the states are uniformly distributed in quasi-energy $\epsilon_b \in [0, \Omega]$, and the total number of states within range $r$ is given by $2^{2r+1}$ we have

$$\int_0^\Omega d\varepsilon \int_0^r dr' \rho(r') = 2^{2r+1} \implies \rho(r) = \frac{4 e^{2r/\zeta_c}}{\zeta_c \Omega}. \tag{40}$$

Consider the $dn = \Omega \rho(r) dr$ states with ranges in the interval $[r, r + dr]$. As they are uniformly distributed over the quasi-energy interval $[0, \Omega]$, the probability that an arbitrarily selected one of them has a quasi-energy in the interval $\varepsilon_\beta^0 \in \varepsilon_\alpha^0 + [-v(r), v(r)]$, and is thus resonant, is given by $2v(r)/\Omega$. It follows that the probability that at least one of these states is resonant with $|\epsilon_a \uparrow\rangle$ is given by

$$q(r) dr = 1 - \left(1 - \frac{2v(r)}{\Omega}\right)^{dn} = 2v(r)\rho(r)dr + \dots, \tag{41}$$

where higher-order corrections in $v(r)/\Omega$ can be dropped for $q(r) \ll 1$. Combining (38), (40) and (41)

$$q(r) = \frac{e^{-r/\xi}}{\lambda}, \tag{42}$$

with $\xi$ as in (39) and the *resonance length* $\lambda$ defined as

$$\lambda := \frac{\zeta_c \Omega}{8J} \approx \frac{\Omega}{J} \gg 1. \tag{43}$$

We expect that $\lambda \gg 1$ as MBL in the RM requires $\Omega \gg J$. Put another way, deep in the MBL phase where $\xi \ll 1$, the probe spin will typically induce resonances of range $r = 0$ (i.e involving only the l-bit $n = 0$ to which it is directly coupled). For stable MBL, the probability of such resonances $q(0) = 1/\lambda$ should be small so that nearest-neighbour resonances are atypical.

### 2.2.3 The probability $p(r)$ that the strongest resonance is at range $r$

The fraction $F(r)$ of states that have not resonated up to range $r$ satisfies the differential equation

$$\frac{\partial F}{\partial r} = -q(r) F(r), \tag{44}$$

with solution

$$F(r) = \exp\left(-\frac{\xi}{\lambda}\left(1 - e^{-r/\xi}\right)\right). \tag{45}$$

The probability $p(r)dr$ that the strongest resonance with the largest matrix element has range in the interval $[r, r + dr]$ is then determined by

$$p(r) = -\frac{\partial F}{\partial r} = \frac{1}{\lambda} \exp\left(-\frac{r}{\xi} - \frac{\xi}{\lambda}\left(1 - e^{-r/\xi}\right)\right). \tag{46}$$

### 2.2.4 The time domain correlator $[C_{zz}(t)]$ and the logarithmically growing light cone front

We now have all the pieces in place to write down the spin-spin correlation function. The strongest resonance for each state is mediated by a matrix element of size $v(r)$ with probability $p(r)$. Plugging this into the pure tone ansatz (34), and treating the disorder average as simply sampling the distribution $p(r)$, we obtain the Floquet RM spectral function

$$[C_{zz}(t)] = [\overline{C}_{zz}] + \int_0^{L/2} dr\, p(r) \cos(v(r)t), \tag{47}$$

where the integral runs over all possible ranges $0 \leq r \leq L/2$, and the infinite time average

$$[\overline{C}_{zz}] := \lim_{T \to \infty} \frac{1}{T} \int_0^T dt \, [C_{zz}(t)] = F(L/2) \tag{48}$$

is simply the probability that a state of the uncoupled system is not resonant with any other state.

We were unable to exactly perform the integral (47). However a crude approximation allows us to extract the asymptotic behaviour in the time domain. At finite time, the dominant contribution to the integral comes from values of $r$ such that $v(r)t$ is small as such terms are always positive. We thus approximate by replacing $\cos(v(r)t) \to \Theta(1 - v(r)t)$ where, $\Theta$ denotes the usual Heaviside step function $\Theta(x > 0) = 1$, $\Theta(x < 0) = 0$. Within this approximation we obtain

$$[C_{zz}(t)] = F(r(t)), \tag{49}$$

where $r(t)$ is obtained by solving $v(r)t = 1$,

$$r(t) = \tfrac{1}{2} \min \left( \zeta_c (1 - \theta_0) \log(Jt), L \right). \tag{50}$$

where we have defined

$$\theta_0 = \frac{\zeta_c}{2\xi + \zeta_c}. \tag{51}$$

The position $r(t)$ has a simple interpretation as the front of a logarithmically growing light cone. Only the cat states formed from l-bits states with $r < r(t)$ contribute to the correlation function at time $t$.

### 2.2.5 The spectral function $[S(\omega)]$

From (47) it is straightforward to obtain the spectral function. For brevity we first recast the matrix element (38) as $v(r) = J e^{-r/(\theta_0 \xi)}$ using (51). Then by inverse Fourier transform of (47)

$$[S(\omega)] = \frac{1}{2} \int_0^{L/2} dr \, \delta(|\omega| - v(r)) p(r) = \frac{\xi \theta_0}{2|\omega|} p\left( \xi \theta_0 \log \left| \frac{J}{\omega} \right| \right). \tag{52}$$

Inserting the calculated form of $p(r)$ (46) into (52) yields

$$[S(\omega)] = \begin{cases} \dfrac{\zeta_c(1 - \theta_0)}{4J\lambda} \cdot \left| \dfrac{\omega}{J} \right|^{-1+\theta_0} \exp\left( -\dfrac{\xi}{\lambda} \left( 1 - \left| \dfrac{\omega}{J} \right|^{\theta_0} \right) \right), & \text{for} \quad \omega_c < |\omega| < J, \\[2ex] [\overline{C}_{zz}] \delta(\omega), & \text{for} \quad \omega_c > |\omega|, \end{cases} \tag{53}$$

in the MBL phase of the Floquet RM. The cutoff scale $\omega_c$ is set by the smallest matrix elements at distance $L/2$,

$$\omega_c = v_{L/2} = J \exp\left( -\frac{L}{2\xi} - \frac{L}{\zeta_c} \right) = \frac{\omega_H}{\lambda} \exp(-L/2\xi), \tag{54}$$

where the Heisenberg frequency $\omega_H := \Omega 2^{-L}$ is set by the typical many-body level spacing.

The high-frequency ($\omega \approx J$) behaviour of $[S(\omega)]$ depends on the microscopic Hamiltonian in the immediate vicinity of the spin, and is thus non-universal. In contrast, the exponent $\theta$ characterising the power-law at low frequency:

$$[S(\omega)] \sim \omega^{-1+\theta} \tag{55}$$

is a consequence of distant resonances which reconfigure large regions of the chain. Thus, as $\zeta \to \zeta_c^-$, we expect $\theta$ to have a universal functional dependence on $|\zeta - \zeta_c|$.

For $L \gg \lambda$ (region II in Fig. 1a), it follows from (53) that

$$\theta = \begin{cases} \theta_0 = \dfrac{\zeta_c}{\zeta_c + 2\xi}, & \zeta < \zeta_c, \\[2mm] \theta_c = \dfrac{\zeta_c}{2\lambda}, & \zeta = \zeta_c. \end{cases} \tag{56}$$

That is, $\theta$ vanishes linearly with $|\zeta - \zeta_c|$ as $\zeta \to \zeta_c^-$, but jumps to a non-universal non-zero value at the transition.

For $L \lesssim \lambda$ (region I in Fig. 1a), $\theta = \theta_c + O(\theta_0)$, so that the exponent is continuously varying. The low-frequency divergence in $[S(\omega)]$ is strongest when $\theta = \theta_c$, we return to this in Sec. 5.2 [3].

Eq. (55) implies that that disorder-averaged correlators exhibit a power-law decay at long times $t \gg J^{-1}$ in the RM MBL phase:

$$[S(\omega)] \sim \omega^{-1+\theta} \iff [C_{zz}(t)] \sim (Jt)^{-\theta}. \tag{57}$$

The decay persists until time $\sim \omega_c^{-1}$, which is exponentially larger than the Heisenberg time $\sim \omega_H^{-1}$. The dynamics at these long time scales are due to the exponentially small (in $L$) fraction of cat states involving re-configurations of l-bits on the scale of the system size $L$.

A fraction of the eigenstates $|\epsilon_a \sigma\rangle$ do not hybridise with any other states despite the coupling with the probe spin to the chain, even as $L \to \infty$. As the probe spin has a well-defined orientation in these states (even upon including perturbative corrections), these states contribute to the infinite-time memory $[\overline{C}_{zz}]$ of the MBL phase.

We defer more detailed discussion of the finite-size behaviour of $[S(\omega)]$ to Sec. 5.

## 2.3 Spectral function of $\sigma_P^z$ in the RM thermal phase $\zeta < \zeta_c$

In the thermal phase, we expect that the off-diagonal matrix elements obey the eigenstate thermalization hypothesis. In particular, the off diagonal matrix elements they do not decay exponentially with range $r$ at large $r$, as assumed by the RM in (12). Consequently, the RM does not apply in this regime.

Despite being generally inapplicable, the early time predictions of the RM are found to hold even in the thermal regime. Specifically, as the probability of resonance $q(r)$ is small for $r \ll |\xi|$, $[S(\omega)]$ exhibits power-law decay (as in (55)) for $J \gg \omega > \omega_\xi$ where,

$$\omega_\xi := v(|\xi|) = J e^{-1/|\theta_0|}. \tag{58}$$

That is, the correlator's dynamics are critical until a time-scale $\sim \omega_\xi^{-1}$. This result is obtained exactly as in the MBL case, with the refinement that, instead of working in a basis of l-bits (which do not exist in the thermal regime), it is necessary to work in a basis of "almost-l-bits" $\tilde{\tau}_n^z$ [73]. These operators have the same properties as l-bits (mutually commuting exponentially localised etc.), but only "almost commute" with the Hamiltonian

$$|[H, \tilde{\tau}_n^z]| \lesssim \omega_\xi. \tag{59}$$

---

[3]We note that the RM predicts that $\theta = \theta_0 < 0$ for $\zeta > \zeta_c$ leading to a stronger divergence than at $\theta = \theta_c$. However, as this prediction hinges on the exponential growth of $q(r)$ on the thermal side for ranges $r < \xi$, this prediction is unphysical and and may be disregarded.

# 3 Hamiltonian resonance model

We describe the computation of the spectral function of the RM with Hamiltonian dynamics. Despite the Hamiltonian case appearing superficially simpler than the Floquet case (as it lacks the additional "ingredient" of a drive frequency) the analysis is more complicated due to the conservation of energy. The associated hydrodynamic mode constrains the late time dynamics, and hence the low frequency behaviour of the spectral function.

For simplicity, we assume that the chain has a single hydrodynamic mode. The analysis is easily generalised to accommodate further conservation laws, such as the spin conservation present in the "standard model of MBL" the Heisenberg model with random $z$-fields.

## 3.1 Set-up

### 3.1.1 Chain Hamiltonian

Consider a strongly disordered static chain with disorder strength $W$ and interaction strength $J$. For specificity, consider the $\Omega \to \infty$ limit of the Floquet model in (4), that is, the Heisenberg model with $O(3)$ random fields

$$H = \frac{J}{2} \sum_n \boldsymbol{\sigma}_n \cdot \boldsymbol{\sigma}_{n+1} + \frac{W}{2} \sum_n \boldsymbol{\nu}_n \cdot \boldsymbol{\sigma}_n. \tag{60}$$

As before, the details of this model will be unimportant except for two key properties: (i) energy is the only conserved extensive quantity at any $W, J$, and (ii) the model is many-body localised for some finite $W \gg J$.

### 3.1.2 The local energy $\epsilon_a$

In addition to its energy eigenvalue $E_a$, each eigenstate $|E_a\rangle$ of $H$ can be assigned a local energy $\epsilon_a(r)$ which can loosely be understood as the expectation value of the Hamiltonian restricted to the sites $n \in [-r, r]$:

$$\epsilon_a(r) \approx \langle E_a | H_{[-r,r]} | E_a \rangle, \tag{61}$$

Here $H_{[-r,r]}$ is the Hamiltonian (60) with the summation restricted to terms acting on the sites $n \in [-r, r]$.

We make this notion sharp with the following definition

$$\epsilon_a(r) = E_a - E_0(a, r) \tag{62}$$

where the energy shift $E_0(a, r)$ is obtained by averaging the energies of the $2^{2r+1}$ states within range $r$ of $|E_a\rangle$

$$E_0(a, r) = \frac{1}{2^{2r+1}} \sum_{b : r_{ab} \leq r} E_b. \tag{63}$$

The local energy has two useful properties. First, for two states $|E_a\rangle$, $|E_b\rangle$ within range $r$, energy differences are preserved exactly

$$E_a - E_b = \epsilon_a(r) - \epsilon_b(r) \iff r_{ab} \leq r. \tag{64}$$

Second, given a state $|E_a\rangle$, the distribution of the local energies $\epsilon_b(r)$ of the states within range $r$ is Gaussian and centred at $\epsilon = 0$. Specifically,

$$\sum_{b : r_{ab} \leq r} \delta(\epsilon - \epsilon_b(r)) \sim \frac{2^{2r+1}}{s_\epsilon(r)\sqrt{2\pi}} \exp\left(-\frac{\epsilon^2}{2s_\epsilon^2(r)}\right) \tag{65}$$

where $\sim$ denotes convergence in distribution at large $r$. Neglecting sub-leading corrections in $J/W$, the width of the Gaussian is given by

$$s_\epsilon(r) = W\sqrt{2r+1}\,. \tag{66}$$

### 3.1.3 Coupling a probe spin to the disordered chain

The Hamiltonian of the chain coupled to a probe spin is given by $\mathcal{H} = \mathcal{H}_0 + \mathcal{H}_1$ with

$$
\begin{aligned}
\mathcal{H}_0 &= H \otimes \mathbb{1} + \frac{h}{2}\mathbb{1} \otimes \sigma_{\mathrm{P}}^z\,, \\
\mathcal{H}_1 &= V \otimes \sigma_{\mathrm{P}}^x\,.
\end{aligned}
\tag{67}
$$

The eigenvectors of $\mathcal{H}$, $\mathcal{H}_0$ and $H$ are denoted $|\mathcal{E}_\alpha\rangle$, $|\mathcal{E}_\alpha^0\rangle$ and $|E_a\rangle$ respectively. These vectors play roles in direct analogy with $|\varepsilon_\alpha\rangle$, $|\varepsilon_\alpha^0\rangle$ and $|\epsilon_a\rangle$ from the Floquet case in Sec. 2. The eigenvectors and corresponding eigenvalues of $H$ and $\mathcal{H}_0$ are related by

$$|\mathcal{E}_\alpha^0\rangle := |E_a\sigma\rangle := |E_a\rangle \otimes |\sigma\rangle\,, \tag{68}$$

$$\mathcal{E}_\alpha^0 := E_a + \tfrac{1}{2}\sigma h\,. \tag{69}$$

Each eigenstate $|E_a,\sigma\rangle$ of $\mathcal{H}_0$ is assigned a local energy

$$e_{(a,\sigma)}(r) = \epsilon_a(r) + \sigma h/2\,. \tag{70}$$

## 3.2 Spectral function of $\sigma_P^z$ in the RM MBL phase $\zeta < \zeta_{\mathrm{c}}$

Our aim is to calculate the disorder averaged infinite temperature spin-spin correlator

$$[C_{zz}(t)] = \frac{1}{\mathcal{D}}\mathrm{tr}\left(\sigma_{\mathrm{P}}^z(t)\sigma_{\mathrm{P}}^z(0)\right) = \int \mathrm{d}t\, e^{-i\omega t}[S(\omega)]\,, \tag{71}$$

for time evolution generated by the Hamiltonian

$$\sigma_{\mathrm{P}}^z(t) = e^{i\mathcal{H}t}\sigma_{\mathrm{P}}^z e^{-i\mathcal{H}t}\,. \tag{72}$$

As in Sec. 2.2, states with resonant partners contribute a pure tone, while states with no resonant partners contribute unity (see (24)), and hence $[S(\omega)]$ follows.

The key difference between the Floquet and Hamiltonian cases stems from the energy dependence of the density of states at range $r$. In the Floquet case, at sufficiently large range $r$, the density of states at range $r$ is independent of quasi-energy, thus all states states have an equal probability of finding a resonance at range $r$. In contrast, in the energy conserving case, states with unusually high/low local energy $e_\alpha(r)$ couple to an atypically small density of states at range $r$. As such these atypical states find resonances at a significantly lower rate (see Fig. 6). We thus adapt the calculation to keep track of the local energy $e_\alpha(r)$ of the states. This leads to a slower decay of $F(r)$, and hence a slower than power law decay of correlations.

### 3.2.1 Identifying resonances

Recall the resonance condition: two states $|E_a \uparrow\rangle$ and $|E_b \downarrow\rangle$ that differ at range $r$ are said to be resonant if

$$|E_a - E_b + h| < |V_{ba}|\,. \tag{73}$$

Using (64), this condition is recast as

$$|e_{(a,\uparrow)}(r) - e_{(b,\downarrow)}(r)| < |V_{ba}|\,. \tag{74}$$

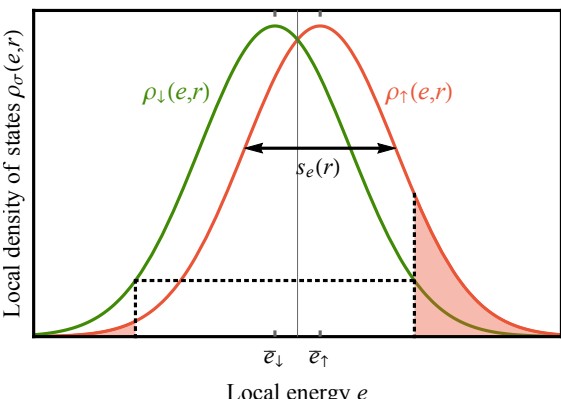

Figure 6: *Local density of states:* the local density of states at range $r$ and energy $e$, $\rho_\sigma(e,r)$, is plotted versus the local energy $e$ for the $\sigma = \uparrow$ (red) and $\sigma = \downarrow$ sectors of the probe spin. These distributions have the same width $s_e(r)$ but are offset from each other due to the probe spin energy $\pm h/2$. The probability of a state in the $\uparrow$ sector finding a resonant partner is proportional to the density of states in the $\downarrow$ sector (see (75)). We illustrate this with an arbitrary cut-off: the $\uparrow$ states at energies $e \notin \overline{e}_\downarrow + [-2s_e(r), 2s_e(r)]$ (red shaded area) have a much reduced probability of resonating versus those in the bulk of the distribution.

### 3.2.2 The probability $q(e,r)$ of finding a resonance at range $r$, and local energy $e$

Define $q_\uparrow(e,r)\big|_{e=e_{(a,\uparrow)}(r)}$, the probability that a state $|E_a \uparrow\rangle$ with finds a resonant partner state $|E_b \downarrow\rangle$ at range $r$. Analogous to the Floquet case, $q_\uparrow(e,r)$ is given by

$$q_\uparrow(e,r) = 2\rho_\downarrow(e,r)v(e,r). \tag{75}$$

where $\rho_\downarrow(e,r)$ is the density of states in the down sector (i.e. the opposite spin sector) at local energy $e = e_{(b,\downarrow)}(r)$ and range $r$, and $v(e,r)$, the characteristic size of matrix elements, coupling states from the two spin sectors at local energies $e$, and range $r$.

Consider the characteristic matrix element $v(e,r)$. To begin with, we neglect the energy dependence of $v$ and assume that the matrix element have the same form as in (38),

$$v(e,r) = Je^{-r/\xi - 2r/\zeta_c}. \tag{76}$$

We later discuss refinements to this approximation.

Next, the density of states $\rho_\sigma(e,r)$ follows from (65),

$$\int_0^r dr' \rho_\sigma(e,r') = \frac{2^{2r+1}}{s_e(r)\sqrt{2\pi}} \exp\left(-\frac{1}{2}\left(\frac{e-\overline{e}_\sigma}{s_e(r)}\right)^2\right). \tag{77}$$

The mean is biased away from zero due to the orientation of the probe spin

$$\overline{e}_\sigma := \tfrac{1}{2}\sigma h \tag{78}$$

and the variance $s_e(r)$ is set by (66). Differentiating (77) and taking the asymptotically dominant behaviour we obtain

$$\rho_\sigma(e,r) \sim \frac{4}{\zeta_c s_e(r)\sqrt{2\pi}} \exp\left(\frac{2r}{\zeta_c} - \frac{1}{2}\left(\frac{e-\overline{e}_\sigma}{s_e(r)}\right)^2\right). \tag{79}$$

Equivalently stated, the asymptotic behaviour of $\rho_\sigma(e,r)$ is dictated by the growth-diffusion equation

$$\frac{\partial \rho_\sigma}{\partial r} = W^2 \frac{\partial^2 \rho_\sigma}{\partial e^2} + \frac{2}{\zeta_c} \rho_\sigma,$$

$$\rho_\sigma\left(e, -\tfrac{1}{2}\right) = \frac{2\sqrt{2}}{\zeta_c} \delta\left(e - \bar{e}_\sigma\right), \tag{80}$$

where the boundary condition is obtained by matching the solutions with (79).

Substituting Eqs. (79) and (76) in (75)

$$q_\uparrow(e,r) \sim \frac{1}{\sqrt{4\lambda r}} \exp\left(-\frac{r}{\xi} - \frac{(e - \bar{e}_\downarrow)^2}{4W^2 r}\right). \tag{81}$$

and similarly for $q_\downarrow(e,r)$. As before $1/\xi = 1/\zeta - 1/\zeta_c$, and the resonance length is defined as,

$$\lambda = \pi \left(\frac{\zeta_c W}{4J}\right)^2 \approx \frac{W^2}{J^2}. \tag{82}$$

The approximation indicates the dropping of an unimportant numerical factor $\pi/(4\log 2)^2 \approx 0.4$. As expected, $q_\sigma(e,r)$ is decaying in $r$ on the localised side ($\xi > 0$), and growing on the thermal side ($\xi < 0$).

### 3.2.3 The probability $p(r)$ that the strongest resonance is at range $r$

The growth diffusion equation (80), which describes the total density of states at local energy $e$ and range $r$, is easily modified to describe the density of states *which have not found a resonant partner* by range $r$. At each range $r$, the hybridisation probability is set by $q_\sigma(e,r)$. We thus obtain:

$$\frac{\partial \rho_\sigma^{\mathrm{u}}}{\partial r} = W^2 \frac{\partial^2 \rho_\sigma^{\mathrm{u}}}{\partial e^2} + \frac{2}{\zeta_c} \rho_\sigma^{\mathrm{u}} - \rho_\sigma^{\mathrm{u}} q_\sigma. \tag{83}$$

Here the superscript 'u' (for unhybridised) distinguishes $\rho_\sigma^{\mathrm{u}}$ from the total density of states $\rho_\sigma$.

We now extract the probability $p(r)$ that a state $|E_a \sigma\rangle$ finds its strongest resonance at a range $r$. Observe that the second term in (83) leads to exponential growth with $r$. Define a distribution that scales out this exponential growth:

$$f_\sigma(e,r) = \frac{\zeta_c}{4\sqrt{2}} e^{-2r/\zeta_c} \rho_\sigma^{\mathrm{u}}(e,r). \tag{84}$$

Substituting in (83), we obtain

$$\frac{\partial f_\sigma}{\partial r} = W^2 \frac{\partial^2 f_\sigma}{\partial e^2} - f_\sigma q_\sigma,$$

$$f_\sigma\left(e, -\tfrac{1}{2}\right) = \delta\left(e - \tfrac{1}{2}\sigma h\right). \tag{85}$$

The substitution (84) has a simple interpretation:

$$F(r) = \int \mathrm{d}e f_\sigma(e,r) = \frac{\int \mathrm{d}e \rho_\sigma^{\mathrm{u}}(e,r)}{\int \mathrm{d}e \rho_\sigma(e,r)} \tag{86}$$

is the fraction of states which have not hybridised by range $r$. Eq. (85) is invariant under the replacements $(e, \sigma) \to (-e, -\sigma)$, by this symmetry $F(r)$ is independent of $\sigma$. It follows that

the probability $p(r)\mathrm{d}r$ that the strongest resonance of a given state is in the interval $[r, r+\mathrm{d}r]$ is given by

$$p(r) = -\frac{\partial F}{\partial r} = \int \mathrm{d}e\, f_\uparrow(e, r) q_\uparrow(e, r). \tag{87}$$

Eq. (87) is the generalisation of the Floquet result (46) to the energy conserving case. Here it is necessary to solve the two-dimensional partial differential equation (85) rather than the simpler one-dimensional ordinary differential equation (44).

What do the solutions of (85) and (87) look like? We discuss two regimes. The first regime in Sec. 3.2.4 is most relevant for the numerically accessible MBL-thermal crossover in Fig. 1b. The second regime of $L, |\xi| \gg \lambda$ determines properties of the Hamiltonian RM in the vicinity of $\zeta = \zeta_c$ as $L \to \infty$ and is discussed in Appendix B.

### 3.2.4 Far from criticality $|\xi| < \lambda$, or small critical systems $L < \lambda < |\xi|$

Neglecting the energy dependence of $q_\sigma(e, r)$,

$$q_\sigma(e, r) \approx \frac{e^{-r/\xi}}{\sqrt{4\lambda r}}. \tag{88}$$

Substituting (88) into (87), we obtain an approximate equation for $F(r)$,

$$\frac{\partial F_1}{\partial r} = -\frac{e^{-r/\xi}}{\sqrt{4\lambda r}} F_1, \tag{89}$$

which we denote as $F_1(r)$ to distinguish it from a true solution to the growth diffusion equations (85) and (86).

Let us justify the approximation above *a posteriori*. For $\xi \gg L$, the solution $F_1(r)$ of (89) decays exponentially on the length scale set by $\lambda$. Thus for $r < \lambda$, the bulk of the weight of the distribution of unhybridised states $f_\sigma(e, r)$ is at typical energies $|e| < s_e(r)$, where the energy dependence of $q_\sigma(e, r)$ can be neglected by making the replacement $q_\sigma(e, r) \to q_\sigma(0, r)$ in (87) to obtain (89). The approximation is thus valid for small critical systems $L < \lambda < |\xi|$ (region I of Fig 1). Far from the crossover on the MBL side $|\xi| < \lambda$, few resonances form after the length scale $\xi$ and $f_\sigma(e, r)$ does not becomes small at typical energies $|e| < s_e(r)$. The bulk of the weight of the distribution of unhybridised states $f_\sigma(e, r)$ is thus at typical energies and the approximation is justified.

On longer length scales $r \gg \lambda$ at $1/\xi = 0$, the weight of $f_\sigma(e, r)$ at typical energies is depleted by the exponential decay. The weight of the distribution is instead concentrated at atypical energies $|e| > s_e(r)$ where the resonance probability $q_\sigma(e, r)$ is much smaller. Appendix B discusses the behaviour at $r \gg \lambda$ in detail.

The solution to the approximated equation (89) is

$$F_1(r) = \begin{cases} \exp\left(-\sqrt{\dfrac{\pi\xi}{4\lambda}}\, \mathrm{Erf}\left(\sqrt{\dfrac{r}{\xi}}\right)\right), & \xi > 0, \\[2em] \exp\left(-\sqrt{-\dfrac{\pi\xi}{4\lambda}}\, \mathrm{Erfi}\left(\sqrt{-\dfrac{r}{\xi}}\right)\right), & \xi < 0, \end{cases} \tag{90}$$

where $\mathrm{Erf}(\cdot)$ and $\mathrm{Erfi}(\cdot)$ are the usual error function and imaginary error function respectively. The correlator then immediately follows

$$[C_{zz}(t)] = F_1(L/2) + \int_0^{L/2} \mathrm{d}r\, p(r) \cos(\nu(r) t). \tag{91}$$

Using (87) and (52), we obtain the desired result:

$$
[S(\omega)] = \begin{cases}
\frac{1}{4J}\sqrt{\frac{\zeta_c(1-\theta_0)}{2\lambda\log|J/\omega|}}\left|\frac{\omega}{J}\right|^{-1+\theta_0}\exp\left(-\sqrt{\frac{\pi\xi}{4\lambda}}\,\mathrm{Erf}\left(\sqrt{\theta_0\log\left|\frac{4J}{\omega}\right|}\right)\right), & \text{for } \xi > 0,\\
& \omega_c < |\omega| < J,\\[2ex]
\frac{1}{4J}\sqrt{\frac{\zeta_c(1-\theta_0)}{2\lambda\log|J/\omega|}}\left|\frac{\omega}{J}\right|^{-1+|\theta_0|}\exp\left(-\sqrt{-\frac{\pi\xi}{4\lambda}}\,\mathrm{Erfi}\left(\sqrt{-\theta_0\log\left|\frac{J}{\omega}\right|}\right)\right), & \text{for } \xi < 0,\\
& \omega_\xi,\omega_c < |\omega|,\\[2ex]
[\overline{C}_{zz}]\delta(\omega), & \text{for } \xi > 0,\\
& \omega_c > |\omega|.
\end{cases}
\tag{92}
$$

The spectral function exhibits the same $\omega^{-1+\theta_0}$ low frequency behaviour as (53) in the Hamiltonian RM MBL phase and at intermediate frequencies in the thermal phase. However, as the localisation length approaches the critical value $\zeta \to \zeta_c$, the correlation length diverges $1/\xi \to 0$, the correlation decay exponent $\theta_0 \to 0^+$, and the correction to the low-frequency $\omega^{-1}$ behaviour of the spectral function is logarithmic rather than power law. We further discuss the logarithmic corrections in Sec. 5.2.2.

## 4 Regime of self consistency of the resonance model

The RM assumes a characteristic range-dependence for the matrix elements $v(r)$ of a local operator $V$ acting at site $n = 0$ (see (38)). The coupling to the probe spin induces hybridisation between the eigenstates of $\mathcal{H}_0$. The reader might thus worry that the off-diagonal matrix elements of a local operator between the hybridised eigenstates is not consistent with the RM assumption in (38). In other words, the distribution of matrix elements after having introduced the probe spin is inconsistent with the distribution we assumed at the beginning.

We address this question in two parts. First, we show that $[S(\omega)] \sim \omega^{-1+\theta_0}$ at low frequencies even if the matrix elements at range $r$ have a generic distribution $p(v|r)$, as opposed to a single value $v(r)$, so long as the aggregate distribution of off-diagonal matrix elements

$$
\varrho(v) = \sum_{r=0}^{L/2} p(v|r)\rho(r)
\tag{93}
$$

is distributed as a power-law in $v$ at small $v$. Thus, we can relax the assumption in (38) to allow for a pre-existing population of resonant cat pairs states, as the matrix elements between such cat pairs and the reference state can differ from $v(r)$.

Next, we imagine perturbing a MBL RM chain, with a given $p(v|r)$, weakly at every site. The local perturbations induce local resonances. When these resonances do not overlap, we argue that the distribution $p(v|r)$ is unaffected at large $r$, and thus that the perturbed chain presents the same statistics of off-diagonal matrix elements $v$ as the unperturbed chain at small $v$. Consequently, the exponent $\theta_0$ that sets the low-frequency divergence of $[S(\omega)]$ is stable to local perturbations.

Specifically, we argue that the resonance model is perturbatively stable, and consequently our conclusions hold, in the regime

$$
\min\left(\frac{L}{2}, |\xi|\right) \ll \sqrt{\lambda},
\tag{94}
$$

in which resonances do not typically overlap. Eq. (94) holds deep in the RM MBL phase as $L \to \infty$ and in region I (see Fig. 1) for sufficiently small systems. Three important conclusions follow:

1. As the RM is self-consistent deep in the MBL phase, the RM predicts and describes a stable MBL phase in the thermodynamic limit.

2. The RM describes the MBL-thermal crossover in short chains, despite being inapplicable at large $L$.

3. The RM describes dynamics in the MBL-thermal crossover at short times as $L \to \infty$, or equivalently on frequency scales:

$$\omega > \omega_{\text{th.}} := \max(\nu(\sqrt{\lambda}), \nu(|\xi|)). \tag{95}$$

## 4.1 Generalised RM with $p(\nu|r)$

Define the *aggregated distribution of off diagonal matrix elements* $\varrho(\nu)$ as the distribution of matrix elements $|V_{ba}|$ that couple two narrow energy windows $E_a \in [E, E+\Delta]$ and $E_b \in [E', E'+\Delta]$ at maximum entropy:

$$\varrho(\nu) := \sum_{ab} \delta(\nu - |V_{ba}|), \tag{96}$$

where $\varrho(\nu)$ and the distribution of matrix elements $p(\nu|r)$ are related by (93). In Secs. 2 and 3, we took the matrix elements at range $r$ to be single valued $p(\nu'|r) = \delta(\nu' - \nu(r))$. In the Floquet case the corresponding aggregated distribution of off diagonal matrix elements at small $\nu$ is

$$\varrho(\nu) = \frac{1}{\Omega} \sum_r N_r \delta(\nu - \nu(r)) \propto \begin{cases} (\nu/J)^{-2+\theta_0}, & \nu < J, \\ 0, & \nu > J, \end{cases} \tag{97}$$

where $N_r = \frac{3}{2} \cdot 4^r$ as in (36), $0 < \theta_0 \le 1$ is defined in (51), and the power law is obtained by coarse-graining over the scale separating the delta functions.

Eq. (53) follows from (97), independent of the precise model $p(\nu|r)$ for the matrix elements at range $r$. Consider the Floquet RM. A change of variables in (44) yields

$$\frac{\mathrm{d}F(\nu)}{\mathrm{d}\nu} = 2F(\nu)\nu \varrho(\nu). \tag{98}$$

The solution

$$F(\nu) = \exp\left( 2 \int_\nu^\infty \mathrm{d}\nu' \, \nu' \varrho(\nu') \right) \tag{99}$$

is the fraction of states which do not have a resonance induced by a matrix element of size $\nu$ or larger. Note that $F(\nu = \infty) = 1$. Similarly we may define

$$p(\nu) := \frac{\partial F}{\partial \nu} = 2\nu \varrho(\nu) \exp\left( 2 \int_\nu^\infty \mathrm{d}\nu' \, \nu' \varrho(\nu') \right), \tag{100}$$

so that $p(\nu)\mathrm{d}\nu$ is the fraction of eigenstates of $\mathcal{H}_0$ whose strongest resonance is due to a matrix element in the range $[\nu, \nu + \mathrm{d}\nu]$. The spectral function is then given by,

$$[S(\omega)] = \tfrac{1}{2}p(|\omega|) + \delta(\omega)F(\nu = 0). \tag{101}$$

Substituting (97), we recover the previously calculated spectral function (53). The calculation presented in Sec. 3 for the Hamiltonian RM can be similarly generalised.

Note that a general model for the matrix elements alters the simple relationship between the localisation length $\zeta$ and the exponent $\theta_0$, and thus leads to an altered critical value of the localisation length $\zeta_c := \zeta|_{\theta_0=0}$.



Figure 7: *Resonances*: The spectral function calculation in the RM is self-consistent if the eigenstates in the RM-MBL are well characterised as l-bit configurations dressed with local resonances. a) A l-bit state dressed with two resonances of range $r = 1$ centred at sites $n = -3$ and $n = 2$. Each resonance is represented by an arc encompassing the patch of rearranged spins. Resonances typically rearrange a patch of size $\xi$ and have density $\xi/\lambda$ (in units of lattice spacing), and thus are well separated for $\xi^2 \ll \lambda$. b) For $\xi^2 \gtrsim \lambda$, these resonances typically overlap forming large resonant patches that destabilise MBL.

## 4.2 Self-consistent and stable localisation

To be self-consistent, the RM must have the same statistical distribution of resonances before and after a local perturbation.

Consider a perturbation $V$ of strength $|V| \approx J$ applied at a single site $n = 0$ (as in Sec. (2.2)). The effect of this perturbation is straightforward: first the eigenstate energies are corrected by the diagonal elements of $V$ (i.e. $E_a \to E_a + V_{aa}$) and second, each state $|E_a\rangle$ finds a resonance at range $r$ (i.e. $|V_{ab}| > |E_a - E_b|$, where $r_{ab} = r$) with probability $q(r) = e^{-r/\xi}/\lambda$. This leads to a pair of resonant 'cat' states

$$\begin{pmatrix} |E'_a\rangle \\ |E'_b\rangle \end{pmatrix} \approx \frac{1}{\sqrt{2}} \begin{pmatrix} 1 & 1 \\ 1 & -1 \end{pmatrix} \begin{pmatrix} |E_a\rangle \\ |E_b\rangle \end{pmatrix}, \tag{102}$$

with corresponding energies $E'_a, E'_b$ and splitting $|E'_a - E'_b| \approx |V_{ab}|$.

We now apply a second perturbation $U$, also of strength $|U| \approx J$, at a site $m$ a finite distance from $n = 0$. Naively, the arguments of Sec. (2.2) imply each such subsequent perturbation causes more long range resonances to develop. However, this is not the case. The matrix element $\langle E'_a | U | E'_b \rangle \approx J e^{-s/\xi}$ where $s = \max(0, m - r_{ab})$ acts to disentangle cat state pairs (102) whose splitting is small $|V_{ab}| \ll J e^{-s/\xi}$. This removes all resonances due to $V$ which are of long range $r_{ab} \gg m/2$. This disentangling of resonances is counterbalanced by the formation of new long range resonances due to the combined action of $U$ and $V$. Their distribution is statistically identical to that induced by a single local perturbation. Specifically, the range of typical resonances remains O($\xi$).

Short range ($r_{ab} \lesssim m/2$) resonances induced by $V$ survive the second perturbation. When the surviving resonances overlap with those induced separately by $U$, the eigenstate entanglement further increases. Specifically, two cat pairs $|E'_a\rangle, |E'_b\rangle$ (102) and $|E'_c\rangle, |E'_d\rangle$ with respective level splittings $|V_{ab}|$ and $|V_{cd}|$ survive if $\langle E'_a | U | E'_b \rangle \lesssim |V_{ab}|$ and $\langle E'_c | U | E'_d \rangle \lesssim |V_{cd}|$ hold. The states $|E'_a\rangle, |E'_c\rangle$ may hybridise if $\langle E'_a | U | E'_c \rangle \gtrsim |E'_a - E'_c|$ yielding $|E''_a\rangle \approx (|E'_a\rangle + |E'_c\rangle)/\sqrt{2}$. In the state $|E''_a\rangle$, a small subsystem in the vicinity of $n = 0$ has entanglement entropy $S \approx 2\log 2$. Similarly two "cats of cats" $|E''_a\rangle$, and $|E''_e\rangle$ may be hybridised by a third perturbation $W$ to form $|E'''_a\rangle \approx (|E''_a\rangle + |E''_e\rangle)/\sqrt{2}$, with entropy $S \approx 4\log 2$. Here we have illustrated the increase of entanglement entropy due to overlapping resonances for the case

$$\langle E''_a | W | E''_e \rangle < \langle E'_a | U | E'_c \rangle < \langle E_a | V | E_b \rangle. \tag{103}$$

The general case is more complex. However we suspect similar increases of the entanglement entropy when resonances overlap.

The merging of local resonances into larger resonant clusters with larger entanglement entropies represents an instability of the "l-bits + local resonances" picture assumed by the RM unless the localisation length is sufficiently short $\xi \ll \sqrt{\lambda}$ (with lengths measured in units of the lattice length). Consider perturbing the RM at every site. At each site, the probability of inducing at least one resonance between the reference state $|E_a\rangle$ and a second state $|E_b\rangle$ is $1 - F(r = \infty) \approx \xi/\lambda$. If the typical spacing between these resonances $\lambda/\xi$ exceeds their typical size $\xi$, then they remain spatially separated. We conclude that for $\xi^2/\lambda \ll 1$ resonances do not merge, and do not alter the asymptotic distribution of matrix elements at low frequencies. The RM is thus self consistent and stable to local perturbations in this regime. This case is depicted in Fig. 7a where the extent of each resonance is indicated by the red arcs. We note that rare states participate in long range resonances $r \gg \xi$; however these do not destabilise the localisation.

Repeating the above arguments for systems of finite-size $L$, we find that resonances occur with density $1 - F(r = L/2) \approx \min(\xi, L/2)/\lambda$ and involve $\min(\xi, L/2)$ sites. This yields the condition (94).

Finally, we note that the RM describes dynamics in the thermodynamically large thermalising phase at short times, or equivalently at frequencies satisfying (95). At these short times, resonances are rare and thus the RM is controlled. As noted in Sec. 2.3, the derivation of $[S(\omega)]$ proceeds through "almost-l-bits" that almost commute with the Hamiltonian.

# 5 RM predictions for finite-size numerics

The RM is self-consistent in short chains

$$L \lesssim 2\sqrt{\lambda} \tag{104}$$

in region I and provides a simple model for the MBL-thermal crossover. Could the RM describe the numerically accessible MBL-thermal finite size crossover? A naive estimate of the resonance length $\lambda$ comes from Eqs. (82) and (43) using numerical and experimentally reported values for the critical frequency or critical disorder strength [6, 63, 74]. This gives $15 \lesssim \lambda \lesssim 50$. Physically, $\lambda$ has to far exceed the lattice scale, as $q(0) = 1/\lambda$ is the probability of a nearest neighbour resonance in the MBL phase. We thus reason that numerically accessible chain lengths $L$ are smaller than or comparable to $2\sqrt{\lambda}$, and that the RM is an analytically tractable model for the numerics.

In what follows, we describe several properties of the RM in short chains that explain numerical observations about the finite-size MBL-thermal crossover. The crossover occurs around the line $|\xi| = L/2$ separating the thermal phase from region I in Fig. 1a). We also explain the numerical observations of Refs. [1] and [2] within the RM. As the RM has a stable MBL phase, we weigh in on the controversy of the existence of MBL in favour of MBL.

## 5.1 Correlation length exponent $\nu = 1$

The thermal-MBL crossover in the resonance model is characterised by a correlation length $|\xi|$:

$$|\xi| \propto |\zeta - \zeta_c|^{-\nu}, \tag{105}$$

which diverges with exponent $\nu = 1$. This value is close to the numerically reported values of $0.77 \leq \nu \leq 1.02$ reported for data collapses of different quantities in Ref [63]. Note that the RM exponent, as well as the numerically reported ones, violate the Harris bound for randomly disordered systems $\nu \geq 2$ [57, 64, 65], as they only capture the pre-asymptotic in $L$ scaling.

## 5.2 Apparent $1/\omega$ divergence of the spectral function

The RM predicts a power-law divergence in $[S(\omega)]$ at low frequencies in the MBL phase and in region I:

$$[S(\omega)] \sim \omega^{-1+\theta}. \tag{106}$$

Above $\sim$ indicates asymptotic equality up to constant factors and log corrections, and $\theta > 0$.

Deep in MBL phase, the following hierarchy of frequency scales hold:

$$\omega_c \ll \omega_H \ll \omega_\xi, \quad 0 < \xi \lesssim L/2 \tag{107}$$

and $[S(\omega)]$ takes the form in (106) for $\omega > \omega_c$ with the exponent $\theta$ given by $\theta_0 > 1$ in (56).

In region I in Fig. (1), $|\xi| \gtrsim L/2$, and the frequency scales are arranged as:

$$\omega_\xi \lesssim \omega_c \sim \omega_H, \quad |\xi| \gtrsim L/2. \tag{108}$$

Below, we show that the low-frequency divergence of $[S(\omega)]$ is strongest in the middle of region I and is given by $[S(\omega)] \propto \omega^{-1}$ up to logarithmic corrections.

Ref. [2] interpreted the apparent $\omega^{-1}$ behaviour as inconsistent with MBL. The RM however predicts this behaviour near the finite-size MBL-thermal crossover in region I and allows for a stable MBL phase.

### 5.2.1 Floquet systems

The exponent $\theta_0$ in (56) vanishes as $|\xi| \to \infty$ in the RM. The strongest low-frequency divergence $[S(\omega)]$ is however not $\sim 1/\omega$ (indeed, as noted in [2] such a strong divergence would violate an elementary sum rule) because the exponential term in (53) modifies the exponent. The RM instead predicts the following spectral function in the middle of region I:

$$[S(\omega)] \sim \omega^{-1+\theta_c}, \quad \omega \gg \omega_c, \omega_H \text{ and } |\xi| \gg \lambda, \tag{109}$$

with $\theta_c = \zeta_c/2\lambda$, as given by (56).

As $\lambda \gg 1$ and $\zeta_c$ is on the lattice scale, we conclude $\theta_c = \zeta_c/2\lambda \ll 1$. The strongest low-frequency divergence in (109) is thus close to $1/\omega$.

Note that (109) implies a power law decay of correlations at late times. Such decay can only be consistent with a logarithmically spreading light cone (50) in the absence of any conserved quantities, such as in a Floquet system.

### 5.2.2 Hamiltonian systems

Hamiltonian systems conserve energy, which results in a logarithmic, rather than power law, correction to $1/\omega$ scaling of $[S(\omega)]$. Specifically, for $|\xi| \gg \lambda \gg L$, we simplify (92) to obtain:

$$[S(\omega)] \sim \frac{1}{\omega\sqrt{\lambda \log|J/\omega|}}, \quad \omega \gg \omega_c, \omega_H. \tag{110}$$

Here $\sim$ indicates equivalence up to an $\omega$ independent pre-factor.

Observe that this decay is not asymptotically consistent with hydrodynamics. The light-cone only grows logarithmically in time in the RM (see Fig. 8), but (110) implies critical correlations that decay faster than $1/\log(Jt)$ as $t \to \infty$,

$$\lim_{\xi \to \infty} [C_{zz}(t)] \sim \exp\left(-\sqrt{\frac{\zeta_c}{2\lambda}\log Jt}\right). \tag{111}$$

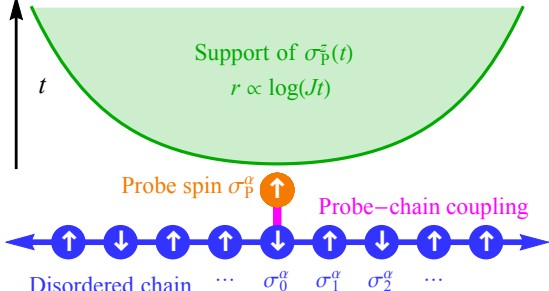

Figure 8: *Logarithmically growing light cone*: the Heisenberg operator $\sigma_{\mathrm{P}}^{z}(t)$ (in (22)) is localised to the probe spin site time $t = 0$. Under time evolution, the support spreads and defines a light cone. After a time $t$, this light-cone has width $r(t) \propto \log J t$ (green).

More careful analysis of the Hamiltonian resonance model finds that below a frequency timescale $\omega_\lambda := \nu(\lambda)$, the decay of $[C_{zz}(t)]$ is dictated by a form

$$[C_{zz}(t)] \sim \frac{1}{\sqrt{\log J t}}, \qquad t > 1/\omega_\lambda \tag{112}$$

consistent with hydrodynamics. We note this corresponds to a time averaged value which goes to zero as $[\overline{C}_{zz}] \sim \xi^{-1/2}$. However, as (112) applies outside of the regime of self-consistency of the resonance model, we relegate further discussion to Appendix B.

### 5.3 Localised finite-size crossover

As the resonance probability is small for $L \ll 2\sqrt{\lambda}$, the RM predicts a localised finite-size crossover (i.e. a localised region I).

First, the time-averaged correlator $[\overline{C}_{zz}]$ is close to unity in both the Floquet and energy conserving cases, and thus retains long-time memory:

$$\lim_{\xi \to \infty} [\overline{C}_{zz}] = \begin{cases} e^{-L/2\lambda}, & \text{(Floquet)}, \\ e^{-\sqrt{L/2\lambda}}, & \text{(Energy conserving)}. \end{cases} \tag{113}$$

Next, the late-time memory implies that small subsystems of the chain have sub-thermal entanglement entropy. This prediction is in agreement with numerical observations in Ref. [58].

Finally, dynamics in the finite size crossover is characterised by a dynamical exponent $z = \infty$ as per the logarithmically growing light cone (see (50) and Fig. 8). The length-energy relationship set by the matrix elements $t \sim \nu(r)^{-1}$ determines the light cone; any l-bits outside the light cone are not entangled with the probe spin. In the thermal phase, we expect that the logarithmic expansion of the light cone crosses over to ballistic or diffusive expansion for $t > \omega_\xi^{-1}$ in Floquet and Hamiltonian systems respectively.

Ref. [75] numerically observed stretched exponential decay of typical spatial correlations in eigenstates in the MBL-thermal crossover region and noted the similarity of their numerical results to that near an infinite-randomness fixed point. Although we do not flesh out the connection between the RM transition and the infinite-randomness transition here, we note that both theories predict $z = \infty$ and logarithmically growing light cones.

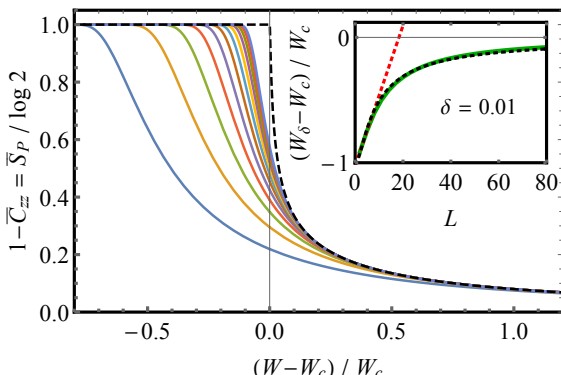

Figure 9: *Drift of the critical disorder strength $W_c(L)$ with L at small sizes:* The main plot shows the probe spin eigenstate averaged entanglement entropy $[S_P]$ predicted by our analysis of the resonance model (calculated using Eqs. (119,92)) as a function of disorder strength for $L = 5, 10, 15, \ldots 60$ (coloured solid lines). The dashed black dashed line indicates the $L \to \infty$ limit. Inset: corresponding values of $W_\delta$ with $\delta = 0.01$ (green solid line) vs $L$, and the corresponding analytic curve from (116) (black dotted line). The red dotted line is a linear fit at small $L$. We see that $W_c(L) \propto L$ at small $L$. Parameters: $1/\xi = \log(W/W_c)$, $W_c = 10$, $\lambda$ as given by (82), and $J = 1$.

## 5.4 Scale-free resonances near the finite-size crossover

In region I (and II), the probability of resonance at range $r$ is scale free

$$\lim_{\xi \to \infty} q(r) = \begin{cases} \dfrac{1}{\lambda}, & \text{(Floquet)}, \\[2mm] \dfrac{1}{\sqrt{r\lambda}}, & \text{(Energy conserving)}, \end{cases} \tag{114}$$

resulting in the formation of resonances on all length scales. This feature of the thermal-MBL crossover in small systems has been observed numerically in Ref. [62].

## 5.5 Linear drift of critical disorder strength with $L$

The RM predicts a ubiquitous feature of small system numerics on disordered chains: that the critical disorder strength increases approximately linearly with $L$. Refs. [1] and [2] argued this drift to be inconsistent with the existence of MBL; the RM however provides an alternative explanation.

The origin of the drift lies in the localised nature of region I. On increasing $1/\zeta$ at small sizes, the chain crosses over from thermal to localised behaviour when the correlation length first exceeds the system size $|\xi| \approx L$ (see Fig. 1). The critical $1/\zeta$ (and equivalently the critical disorder strength) thus increase with $L$.

This drift can be quantified: let $W_\delta(L)$ denote the disorder strength at which the time-averaged correlator $[\overline{C}_{zz}]$ deviates from its value in the infinite temperature Gibbs ensemble by some small amount $\delta$,

$$[\overline{C}_{zz}(W_\delta)] = \delta \ll 1. \tag{115}$$

For the Hamiltonian RM, algebraic manipulation of (90) with $1/\xi = \log(W/W_c)$ yields:

$$W_\delta(L) \approx W_c e^{-\ell_\delta/(L+1)}. \tag{116}$$

for some $\delta$-dependent constant $\ell_\delta$. Over a regime of sufficiently small $L$, this function is approximately linearly increasing with $L$ (see Appendix D for derivation).

More generally the linear growth of $W_\delta$ follows from Taylor expanding $\xi$ near $W = W_\delta$. Precisely, if we identify $\xi(W_\delta(L)) \propto L$, (for some $\delta$-dependent constant of proportionality), and consider the taylor expansion

$$\xi(W) = \xi(W_\delta(L)) + (W - W_\delta(L))\xi'(W_\delta(L)) + \dots \tag{117}$$

about the point $W = W_\delta(L + \Delta L)$ we obtain

$$\Delta W_\delta := W_\delta(L + \Delta L) - W_\delta(L) \propto \frac{\Delta L}{\xi'(W_\delta(L))}. \tag{118}$$

Eq. (118) and the linear-in-$L$ drift of the critical point follow provided $W$ is sufficiently far from the transition that i) the Taylor expansion is valid (i.e. $|W - W_\delta(L)| < |W_\delta(L) - W_c|$) and ii) that $\xi'(W_\delta(L))$ is slowly varying in $L$.

The drift in the critical point has been observed in various statistics across many studies []. As an example we consider the spectrally averaged spin (or p-bit) eigenstate entanglement entropy $[S_P]$. This quantity is finite in the MBL phase $[S_P] = O(1)$ (tending to zero at strong disorder), whereas in the thermal phase $[S_P] = \log 2$ up to corrections which are exponentially small in $L$. Within the resonance model, $[S_P]$ is given by

$$[S_P] = \log 2 \left( 1 - [\overline{C}_{zz}] \right). \tag{119}$$

In Fig. 9 we plot $[S_P]$ for the Hamiltonian RM as a function of the re-scaled disorder strength (using $1/\xi = \log(W/W_c)$). The probe spin entropy is maximal in the cat states, and is zero is the fraction $[\overline{C}_{zz}] = F(L/2)$ of states that do not resonate. The inset confirms that the deviation $(W_\delta - W_c)$ increases linearly with $L$ at small $L$, before converging to zero from below at large $L$.

A similar analysis in the Floquet RM predicts a linear drift of the critical frequency at which localisation sets in with $L$ for fixed disorder strength.

## 5.6 Exponential increase of the Thouless time with disorder strength

Refs. [1] and [2] numerically studied the scaling of the Thouless time with disorder strength in the thermalising phase. The Thouless time is defined as the time-scale above which random matrices govern quantum dynamics in chaotic systems, or equivalently as the inverse of the energy scale below which the random matrices govern eigenstate properties. Through a detailed study of the spectral form factor and $[S(\omega)]$, Refs. [1] and [2] argued that the inverse of the Thouless time $\omega_{\text{Th.}}$ exponentially decreases with disorder strength:

$$\omega_{\text{Th.}} \propto e^{-cW/J}. \tag{120}$$

Should this behaviour continue asymptotically as $L \to \infty$, then the numerically observed MBL-thermal crossover is simply a finite-size effect caused by $\omega_{\text{Th.}}$ becoming smaller than the Heisenberg time $\omega_{\text{H}}^{-1}$. That is, the observed localisation is simply a consequence of the small sizes accessible to exact numerics.

The RM provides an alternate explanation for (120) while allowing for a MBL phase. In a diffusive system, the Thouless time is set by the time taken by a localised packet of energy to spread over the system. For diffusion constant $D$, thus $\omega_{\text{th.}} = D/L^2$. As the packet takes time $\omega_\xi^{-1}$ to spread a distance $\xi$, $D = \omega_\xi \xi^2$. Combining these estimates

$$\omega_{\text{Th.}} = \frac{D}{L^2} = \frac{\omega_\xi \xi^2}{L^2} \approx \frac{J\xi^2}{L^2} e^{-2|\xi|/\zeta_c}, \tag{121}$$

where $\approx$ indicates the dropping of an $O(1)$ factor.

Next, consider the correlation length $\xi(W)$. It is a smooth function of the disorder strength $W$ and diverges at the critical disorder $W_c$ defined by $\zeta = \zeta_c$. As discussed in Sec. 5.5, the crossover from spectrally averaged statistics being close to their thermal values, to close to their localised values occurs at disorder strength $W_\delta$, a much weaker disorder strength than $W_c$ in small systems sizes. We may thus Taylor expand $\xi$ near $W = W_\delta$ (as in (117)) from which the exponential dependence of the Thouless time on the disorder strength $W$ of (120) follows.

## 5.7 Apparent sub-diffusion in the RM thermal phase

Eqs. (53) and (90) predict a continuously varying exponent for the spectral function $[S(\omega)] \sim \omega^{-1+\theta}$ above a threshold frequency scale $\omega_\xi$ in the thermal phase. The RM thus explains the apparent sub-diffusion (as measured by the dynamic exponent $1/\theta$) reported in several studies [66–70] without any reference to rare regions, and indeed predicts such apparent sub-diffusive behaviour even in Floquet systems without any conservation laws. This prediction of the RM may resolve a mystery about the absence of broad distributions of the conductivity (across disorder realisations) that are expected in a sub-diffusive regime characterised by weak links [55, 56].

We note that Ref. [76] (in the supplementary material) previously speculated that rare resonances may lead to apparent sub-diffusive behaviour in the thermal phase.

## 5.8 Exponentially enhanced sensitivity to eigenstates or 'maximal chaos'

The fidelity susceptibility $\chi_a$ measures the sensitivity of an eigenstate $|E_a\rangle$ to perturbation by a local operator $U$. It is defined as

$$\chi_a = \sum_{b \neq a} \left| \frac{\langle E_b | U | E_a \rangle}{E_b - E_a} \right|^2 . \tag{122}$$

The mean of the logarithm of $\chi$ (defined as the average of $\log \chi_a$ across infinite temperature eigenstates and disorder realisations) shows the following scaling with $L$:

$$[\log \chi] \sim \begin{cases} L \cdot \log 2, & \text{thermal}, \\ L^0, & \text{MBL}. \end{cases} \tag{123}$$

Ref. [2] made two observations about the distribution of $\log \chi_a$ at numerically accessible sizes. First, there is a regime of *maximal chaos* separating the thermalising and MBL regimes in which

$$[\log \chi] \sim L \cdot 2 \log 2, \quad (\text{"maximal chaos"}). \tag{124}$$

Second, the tails of the distribution in the putative MBL regime (in which $[\log \chi]$ saturates) are fatter than expected from a Poisson distribution. The authors explained both observations through the exponential enhancement of matrix elements between eigenstates with energy differences comparable to the many-body level spacing, and concluded that such enhancement is inconsistent with MBL.

The RM explains both observations in Ref. [2] assuming a thermodynamic MBL phase. Consider a pair of resonant cat states $|E'_{a,b}\rangle = (|E_a\rangle \pm |E_b\rangle)/\sqrt{2}$ involving the re-arrangement of l-bits at range $r = L/2$ and splitting comparable to or less than the many-body level spacing.

A generic local perturbation $U$ will couple these states as $\langle E'_a|U|E'_b\rangle = O(|U|)$ [4]. Consequently, their fidelity susceptibility is very large, increasing as $\sim 2^{2L}$.

In the numerically accessible MBL-thermal crossover, a finite fraction $q(L/2)\Delta L$ of the eigenstates are involved in resonances with range between $L$ and $L + \Delta L$ and splitting comparable to the many-body level spacing. The RM thus predicts maximum chaos (124) at the finite-size crossover. More precisely, in regions I and II of the Floquet RM

$$[\log \chi] = \int_0^L s(r)\log\left(|U|^2\rho^2(r)\right) = L\left(2\log 2 + O(\lambda/L)\right),\tag{125}$$

where $\rho(r)$ sets the typical inverse level spacing for a resonance at range $r$, and $s(r) = q(r)\exp(-\int_r^{L/2} q(r')\mathrm{d}r')$, is the probability that the longest range resonance for a given state is at range $r$. Thus, maximum chaos is approached as $L$ becomes closer to $\lambda$.

In the RM MBL phase, the fraction of states involved in system-wide resonances $q(L/2)$ is exponentially small in $L$. These states thus do not contribute to $[\log \chi]$, which is independent of $L$. Nevertheless, these rare states lead to increased weight in the tail of the distribution of $\log \chi$. This explains the second observation of Ref. [2].

## 5.9 Absence of a cut-off at the Heisenberg time in the MBL phase

We find that the dynamics in the MBL phase are not cut-off by the Heisenberg time $t_\mathrm{H} \sim \omega_\mathrm{H}^{-1} \sim J^{-1}2^L$. Instead, the RM is cut-off by an exponentially larger in $L$ time-scale set by $\omega_\mathrm{c}^{-1}$:

$$\omega_\mathrm{c} = v(L/2) = \omega_\mathrm{H}\mathrm{e}^{-L/2\xi}.\tag{126}$$

The dynamics on the time-scales $t \gg \omega_\mathrm{H}^{-1}$ are due to the rare cat states with energy splittings that are smaller than the typical level spacing.

The existence of a timescale longer than the Heisenberg time $t_\mathrm{H}$ contradicts commonly held lore that at $t_\mathrm{H}$ the system "realises" that it is finite, the discreteness of the spectrum is resolved, the dynamics becomes quasi-periodic, and thus there cannot be physically meaningful dynamics beyond $t_\mathrm{H}$. This lore neglects that in the localised phase all local operators have discrete (i.e. pure-point) spectra even before $t_\mathrm{H}$, so there is nothing to "realise" at $t_\mathrm{H}$.

## 5.10 A simple numerical stability criterion for MBL

Following the discussion in Sec. 4.2, MBL requires that the expected number of resonances induced by a local perturbation $V$ in a typical eigenstate of the chain is much smaller than unity:

$$\int_0^\infty \mathrm{d}r\, q(r) \ll 1.\tag{127}$$

Using the tools developed in Sec. 4.1, we can re-write the above criterion in-terms of the aggregated distribution $\varrho(v)$ of off-diagonal matrix elements of $V$:

$$\int_0^\infty \mathrm{d}v\, v\varrho(v) = \rho\bar{v} \ll 1.\tag{128}$$

Here $\rho$ is the many body density of states in some small mid spectrum window of width $\Delta$, and

$$\bar{v} = \frac{1}{\Delta\rho}\sum_b |V_{ba}|\tag{129}$$

---

[4]To see this note that if $U = \tau_n^z$ on a site $n$ in which $\tau_{an} \neq \tau_{bn}$, then $U$ has an order one matrix element between the two cat states (and similarly for any string of $\tau_n^z$ with an odd number of such terms). $\langle E'_a|U|E'_b\rangle = O(|U|)$ then follows as a generic local operator $U$ has $O(|U|)$ overlap onto such terms

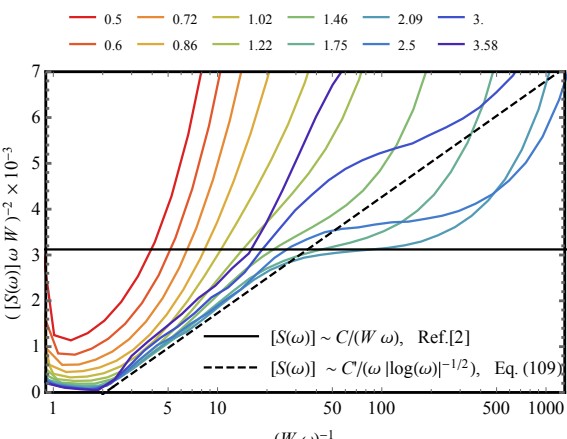

Figure 10: *Spectral function data from Ref. [2]*: Disorder averaged spectral function data for the random XXZ model from Fig. 2a of Ref. [2] (same colour scheme). Different series correspond to different disorder strengths $W$ (legend above). Here we plot $([S(\omega)]\omega W)^{-2}$ as a function of $(W\omega)^{-1}$ so that the pure $1/\omega$ divergence predicted by Ref. [2] appears as a horizontal line (black solid, $C = 0.0179$) whereas the form predicted in this work, (110), appears as line of constant gradient (black dashed). Agreement with (110) is seen for 1.4 decades for $(W\omega)^{-1} \in [1.7, 40]$.

is the mean matrix element in the same window for a mid-spectrum state $a$.

Eq. (129) provides a simple numerically tractable criterion for MBL. As $L \to \infty$, the quantity $\rho\bar{v}$ grows exponentially with $L$ in a thermalising phase that satisfies the eigenstate thermalization hypothesis, but saturates in a MBL phase:

$$\rho\bar{v} \propto 2^{L/2} \text{ (thermal)}, \quad \rho\bar{v} = \text{cons.} \ll 1 \text{ (MBL)}. \tag{130}$$

Note that (129) makes no reference to a l-bit basis. When $\varrho(v) \propto v^{-2+\theta_0}$ at small $v$, the stability criterion implies that $0 \le \theta_0 < 1$ for MBL.

Eq. (129) generalises the stability criterion to thermalising avalanches introduced in Ref. [48]. Ref. [48] studied the stability of a MBL system composed of l-bits to a thermalising inclusion, and argued that $\zeta$ (the length scale controlling the localisation of a physical spin operator in the l-bit basis) must be smaller than $\zeta_c = 1/\log 2$. Re-writing the avalanche criterion in terms of properties of off-diagonal matrix elements, we obtain (129) with no reference to either rare regions or to l-bits.

## 6 Discussion

We have presented the RM, a model of the finite-size MBL-thermal crossover in which the localised phase is destabilised by many-body resonances, rather than rare low-disorder regions. The RM is consistent with a stable MBL phase, and reproduces several numerically observed features of the MBL-thermal finite-size crossover, including the controversial observations of Refs. [1,2].

Fig. 10 re-plots the $[S(\omega)]$ data in Fig. 2 of Ref. [2]. The plot shows the frequency dependence of $[S(\omega)]$ at several disorder strengths $0.5 \le W \le 2.5$ in the putative thermalising phase of the disordered spin-$\frac{1}{2}$ XXZ chain. Ref. [2] argued that the data is consistent with the scaling law $[S(\omega)] \sim C/(W\omega)$ (black horizontal line) over an increasing range of frequencies. We instead argue that the data is consistent with the scaling law predicted by the Hamiltonian

RM with a logarithmic correction (dashed black line). Indeed, the curves for $W \gtrsim 1$ align with the RM prediction over $\approx 1.4$ decades in frequency, while evidence of the plateau predicted by Ref. [2] is visible only in two of the curves with $W \approx 1.5, 1.75$, and over less than a decade in frequency. The behaviour of the curves with $W \approx 1.5, 1.75$ is however noteworthy, and not immediately explained by the RM. To settle the debate between the two scaling predictions requires more systematic numerical investigation of the effects of system size on the curves in Fig. 10. Specifically, numerics at larger $L$ should reveal which of the two regimes (the linear growth or the plateau) expands with increasing $L$.

The RM makes several numerically testable predictions about Floquet and quasi-periodically modulated spin chains. First, Sec. 5 applies without alteration to the quasi-periodic case. Second, the exponent $\theta_c$ controlling the strongest low-frequency divergence of the spectral function in region I in the Floquet case is non-universal and non-zero, in contrast to the Hamiltonian RM with $\theta_c \to 0^+$. Third, Floquet systems on the thermalising side of the finite-size crossover would also exhibit apparent sub-diffusive scaling in their spectral functions. The origin of this apparent sub-diffusion is the formation of many-body resonances on length scales shorter than $\xi$. Fourth, irrespective of the type of disorder or the number of conservation laws, we predict logarithmically growing light cones in the thermalising phase for $t \lesssim \omega_\xi^{-1}$. Finally, observables conditioned on the formation of resonances could detect the MBL-region I crossover in Fig. 1a.

Eq. (130) offers a new numerical criterion to differentiate localised and thermalising systems. Analogous to the $\mathcal{G}$ parameter in Ref. [77] and the typical fidelity susceptibility [2], $\rho\bar{\nu}$ is exponentially larger in $L$ in the thermalising phase as compared to the MBL phase. Preliminary work on a disordered Ising model suggests that (130) bounds the transition out of the localised phase to larger disorder strengths than other standard criteria based on energy level statistics or eigenstate entanglement entropies.

Future work could explore the RM along several axes. The first is to establish whether the distribution of sample conductivities (across disorder realisations) predicted by the RM is consistent with the observations of Ref. [55]. This would add further evidence to the claim that many-body resonances, and not rare regions, give rise to the apparent sub-diffusion observed in numerical studies.

The second is to compare the eigenstate correlations predicted by the Hamiltonian RM to those from the Anderson model on the random regular graph (RRG) [78, 79]. The RRG Anderson transition is believed to model the MBL-thermal transition if one identifies each site of the RRG with a computational basis state of a disordered spin chain [80]. Using Mott-type resonance arguments similar to those of Sec. 3, Ref. [78] recently argued that in the RRG localized phase, the correlator $[\text{tr}(\Pi_n(t)\Pi_n(0))]$ (where $\Pi_n(t)$ is the time evolved single site projector onto the site $n$) has a Fourier spectrum $\beta(\omega)$ which diverges as a power law as $\omega \to 0$. Identifying each $\Pi_n$ with $|E_a\sigma\rangle\langle E_a\sigma|$, a product state of the probe spin and the disordered chain, the RM predicts that $\beta(\omega)$ diverges exactly as $[S(\omega)]$ (27). The reconcilation of the RM with the RRG is however less apparent in the thermal phase, where the latter predicts a correlation length that diverges with a different exponent than in the RM.

The third is to attempt an extension of the RM to the asymptotic limit in systems with correlated disorder. The RM neglects the effects of rare low-disorder regions; these regions dictate the asymptotic transition in randomly disordered systems [45, 46, 57, 61, 64, 81–86]. Contrarily, in MBL chains with quasiperiodic [87–89] or sufficiently hyperuniform [90] disorder, as there are no such rare regions [57, 91, 92], MBL may be destabilised by many-body resonances even in the thermodynamic limit.

# Acknowledgements

We are grateful to S. Gopalakrishnan, D. Huse, A. Polkovnikov, A. Scardicchio, and D. Sels for insightful comments and useful discussions, to P. Krapivsky for insight into the treatment and regimes of (83), and to C.R. Laumann and V. Khemani for detailed comments on a draft of the manuscript. We are additionally grateful to D. Sels for providing the data of Fig 2. from Ref. [2], here plotted in Fig. 10.

**Funding information** P.C. is supported by the NSF STC "Center for Integrated Quantum Materials" under Cooperative Agreement No. DMR-1231319. A.C. work is supported by NSF DMR-1813499.

# A  Multiple and imperfect resonances in the Resonance Model

## A.1  Imperfect cat states

In Sec. 2.2.1, we assume that pairs of resonant eigenstates of $\mathcal{H}_0$ form perfect cat states with equal weights,

$$|\varepsilon_{\alpha,\beta}\rangle = \frac{1}{\sqrt{2}}\big(|\epsilon_a \uparrow\rangle \pm |\epsilon_b \downarrow\rangle\big). \tag{131}$$

Their contribution to $[S(\omega)]$ is thus pure tone with no weight at zero frequency,

$$\langle \epsilon_a \sigma|\sigma_P^z(t)\sigma_P^z(0)|\epsilon_a \sigma\rangle = \cos(|V_{ba}|t). \tag{132}$$

A more refined ansatz for the hybridised states would incorporate the resonance parameter $g_{ba}$ and lead to imperfect cat states:

$$|\varepsilon_{\alpha,\beta}\rangle = \sqrt{p}|\epsilon_a \uparrow\rangle + \sqrt{1-p}\,e^{i\phi}|\epsilon_b \downarrow\rangle. \tag{133}$$

Above, $p \approx 1/2 + O(g_{ba}^{-1})$. Imperfect cat states contribute delta function peaks at $\omega = 0$ and $\omega = \omega_{a\uparrow} \approx |V_{ba}| + O(|V_{ba}|g_{ba}^{-2})$

$$\langle \epsilon_a \sigma|\sigma_P^z(t)\sigma_P^z(0)|\epsilon_a \sigma\rangle = (1-2p)^2 + 4p(1-p)\cos(\omega_{a\uparrow}t). \tag{134}$$

Accounting for the distribution of $g_{ba}$ in (25) corrects $\lambda$, the weight at zero frequency and the exact form of $[S(\omega)]$. However, it does change universal features, such as the vanishing of the exponent $\theta$ with $|\zeta - \zeta_c|$ and the exponential decay in $r$ of $F(r)$, the weight at zero frequency after all range $r' \leq r$ processes have been accounted for, as per (45).

## A.2  Multiple resonances

Suppose an eigenstate $|\epsilon_a, \uparrow\rangle$ is resonant with multiple other eigenstates of $\mathcal{H}_0$. Here we argue that the strongest resonance (defined by (35)) sets the frequency of oscillation of $\langle \epsilon_a \uparrow|\sigma_P^z(t)\sigma_P^z(0)|\epsilon_a \uparrow\rangle$.

Consider the case of two resonances at different ranges. Let $|\varepsilon_\alpha\rangle = \frac{1}{\sqrt{2}}(|\epsilon_a \uparrow\rangle + |\epsilon_b \downarrow\rangle)$ denote the cat state resulting from the strongest resonance (at the shorter range). Suppose that $|\varepsilon_\alpha\rangle$ is now resonant with another state $|\epsilon_c \downarrow\rangle$ at larger range with some matrix element

$$\langle \varepsilon_\alpha|V|\epsilon_c \downarrow\rangle = V_{\alpha c} := \frac{1}{\sqrt{2}}(V_{ac} + V_{bc}). \tag{135}$$

This matrix element is much smaller than $|V_{ba}|$ as $|V_{ac}|, |V_{bc}| \ll |V_{ba}|$. Treating this resonance within degenerate perturbation theory splits the peak at $\omega = |V_{ba}|$ into two peaks at

$\omega = |V_{ba}| \pm |V_{\alpha c}|$. As this further splitting is small, we neglect it and assume that the spectral weight remains sharply peaked around $\omega = |V_{ba}|$.

In the time domain this statement is as follows: an initial state $|\epsilon_a \uparrow\rangle$ oscillates between $|\epsilon_a \uparrow\rangle$ and $|\epsilon_b \downarrow\rangle$ on a time scale $|V_{ba}|^{-1}$ and tunnels into the state $|\epsilon_c \downarrow\rangle$ on the much longer timescale $|V_{\alpha c}|^{-1}$.

We generalise the above argument to many-resonance case. Suppose $|\varepsilon_\alpha\rangle$ has a resonance meditated by a matrix element $|V_{\alpha c}|$, which leads to hybridised states

$$|\varepsilon'_{\alpha\pm}\rangle = \frac{1}{\sqrt{2}}\left(|\varepsilon_\alpha\rangle \pm |\epsilon_c \downarrow\rangle\right). \tag{136}$$

Take one of these states $|\varepsilon'_{\alpha+}\rangle$. Suppose this state has a longer-range resonance mediated by a matrix element $|V'_{\alpha d}|$. We obtain two new cat states. Suppose one of these two cat states $|\varepsilon''_{\alpha+}\rangle$ has an even longer-range resonance mediated by $|V''_{\alpha e}|$ and so on. The initial peak at $\omega_{a\uparrow} = |V_{ba}|$ splits into several peaks at

$$\omega = |V_{ba}| - |V_{\alpha c}|, \ |V_{ba}| + |V_{\alpha c}| - |V'_{\alpha d}|, \ |V_{ba}| + |V_{\alpha c}| + |V'_{\alpha d}| \pm |V''_{\alpha e}| \ldots . \tag{137}$$

An analogous procedure splits each of the peaks with a minus sign in the RHS above into many sub-peaks.

To show that such shift $\Delta\omega$ remain unimportant we calculate the root-mean-square size shift $\overline{\Delta\omega^2}$ as show that $\overline{\Delta\omega^2} \ll \omega_{a\uparrow}$. To do this we first note that the matrix elements $v(r)'$ connecting an already hybridised state to other unhybridised states at range $r$ are a factor $\sqrt{2}$ smaller

$$v'(r) = \frac{1}{\sqrt{2}}v(r), \tag{138}$$

where as the density of states is twice as large

$$\rho'(r) = 2\rho(r), \tag{139}$$

yielding a probability of hybridising at range $r$ of

$$q'(r) = \sqrt{2}q(r). \tag{140}$$

Thus, supposing that the initial resonance is at a range $r$ (i.e. that $\omega_{a\uparrow} = v(r)$) we find

$$\Delta\omega = \sum_{r'=r+1}^{\infty} v'(r')X(r') \tag{141}$$

where $X(r)$ is a random variable which takes values $X(r) = 1, -1, 0$ with probabilities $q'(r)/2$, $q'(r)/2, 1-q'(r)$ respectively. Thus $\Delta\omega$ has mean $\overline{\Delta\omega} = 0$ and, measured in units of the initial resonant frequency $\omega_{a\uparrow}$, has variance

$$\frac{\overline{\Delta\omega^2}}{\omega_{a\uparrow}^2} = \int_{r+1}^{\infty} ds\, q'(s)\left(\frac{v'(s)}{v(r)}\right)^2 = \frac{e^{-(3+r)/\xi}}{16\sqrt{2}\lambda(4/\zeta_c + 3/\xi)}. \tag{142}$$

On the localised half of the phase diagram ($\xi > 0$) this quantity is exponentially decaying in $r$, indicating this approximation scheme is asymptotically improving at low frequencies. In the crossover region it is bounded by its critical value, which is much smaller than unity

$$\frac{\overline{\Delta\omega^2}}{\omega_{a\uparrow}^2} \approx \frac{\zeta_c}{64\sqrt{2}\lambda} \ll 1, \tag{143}$$

and so does not alter the asymptotic form of the spectral function $[S(\omega)]$, whereas on the thermal this approximation breaks down only for $r > \xi$, outside the regime of validity of our calculation.

# B The spectral function $[S(\omega)]$ in the Hamiltonian RM for large systems in the vicinity of the MBL transition: $L, |\xi| > \lambda$

In this regime hydrodynamic constraints become important. These constraints highlight the limitations of the approximation made in (89), as $F_1$ predicts unphysical behaviour. Specifically

$$\lim_{\xi \to \infty} F_1(r) = e^{-\sqrt{r/\lambda}}, \tag{144}$$

which using $[C_{zz}(t)] = F(r(t))$ (49), and the logarithmically growing light cone $r(t) \propto \log t$ implies that the correlations decay as a stretched exponential in $\log t$. This decay is slower than any power law, but much faster than the maximum possible decay rate permitted by energy conservation of

$$[C_{zz}(t)] \propto \frac{1}{r(t)} \propto \frac{1}{\log t}. \tag{145}$$

This maximum rate follows as the $z$-field on the probe spin $\sigma_{\mathrm{P}}^z$ has overlap with the Hamiltonian $\mathrm{tr}\left(\sigma_{\mathrm{P}}^z \mathcal{H}\right) = W$, and any initial energy on the probe spin cannot have spread further than the light cone front $r(t)$.

In order to address this inconsistency we turn to a more careful treatment of Eqs. (85) and (87). By direct numerical integration (see Appendix C.1) we find that the stretched exponential decay is cut-off at $r \gtrsim \lambda$ by an asymptotic decay $F(r) \sim r^{-2}$, implying a decay $[C_{zz}(t)] \sim \log^{-2} t$. This decay is still too fast to be consistent with hydrodynamics, however, the weakness of this violation means there are many small corrections which yield a late time dynamical regime consistent with hydrodynamics. For example, a sub leading power law in $r$ on the matrix elements $v(e, r)$ will suffice. However, here we explore the effect of energy dependency of the matrix elements.

Instead of the energy independent form for the matrix elements (76), we now consider

$$v(e, r) = J \exp\left(-\frac{r}{\tilde{\zeta}(e/r)} - \frac{r}{\zeta_{\mathrm{c}}}\right). \tag{146}$$

where we now allow the localisation length to vary as a function of the energy density $e/r$ of the patch of the system which must be rearranged to relate the two states $|E_a \uparrow\rangle$ and $|E_b \downarrow\rangle$ (As we are interested only in behaviour at asymptotically large $r$, we consider these states to be at the same energy density, despite their energy difference of $\pm W$ due to the probe spin). We consider only the leading order dependence on energy density of the localisation length

$$\frac{1}{\tilde{\zeta}(e/r)} = \frac{1}{\zeta}\left(1 + \frac{e}{r\eta} + \frac{e^2}{r^2\mu^2} + \dots\right), \tag{147}$$

where $\zeta$ is the localisation length at maximum entropy, the constant energy densities $\mu, \eta$ determine scales over which $\zeta$ varies, and we have suppressed higher powers of $e/r$. We will assume $\eta = \infty$ as the statistical symmetry of the model implies $\tilde{\zeta}$ should be an even function, and $\mu$ positive and finite. This corresponds to a localisation length which is shorter away from maximum entropy.

The energy dependence of the matrix elements then alters the form of $q_\sigma(e, r)$:

$$q_\sigma(e, r) \sim \frac{1}{\sqrt{4\lambda r}} \exp\left(-\frac{r}{\xi} - \frac{e^2}{\zeta r\mu^2} - \frac{(e + \bar{e}_\sigma)^2}{4W^2 r}\right). \tag{148}$$

For $\mu$ positive and finite $q_\sigma(e, r)$ is asymptotically narrower than $\rho_\sigma(e, r)$ at large $r$, we can

extract the asymptotic behaviour of $f_\sigma$ by replacing $q_\sigma(e,r)$ with a delta function

$$\frac{\partial f_\sigma}{\partial r} = W^2 \frac{\partial^2 f_\sigma}{\partial e^2} - \gamma \delta(\epsilon + \tfrac{1}{2}\sigma W) f_\sigma,$$

$$f_\sigma\left(e, -\tfrac{1}{2}\right) = \delta\left(e - \tfrac{1}{2}\sigma W\right),$$

(149)

where $\gamma = \int \mathrm{d}e\, q_\sigma(e,r)$ is an $r$ independent constant at the critical point. Solving (149) (see Appendix (C.2)) we find asymptotic decay

$$F(r) = \int \mathrm{d}e\, f_\sigma(e,r) \sim \frac{1}{\sqrt{r}},$$

(150)

where here $\sim$ indicates asymptotic equality up to an overall constant. This yields

$$[C_{zz}(t)] \sim \log^{-1/2} J t,$$

(151)

$$[S(\omega)] \sim |\omega|^{-1} \log^{-3/2} |J/\omega|,$$

(152)

consistent with hydrodynamic restrictions.

## C  Solutions to the loss-diffusion (85)

In this appendix we consider the loss-diffusion equation (85)

$$\frac{\partial f_\sigma}{\partial r} = W^2 \frac{\partial^2 f_\sigma}{\partial e^2} - f_\sigma q_\sigma,$$

$$f_\sigma\left(e, -\tfrac{1}{2}\right) = \delta\left(e - \tfrac{1}{2}\sigma W\right).$$

(153)

We study two regimes:

- We first study the critical dynamics ($\zeta = \zeta_c$) with energy independent matrix elements ($v$ a function of $r$ only). We show that the asymptotic decay of $F(r) = \int \mathrm{d}e\, f_\sigma(e,r)$ is given by $F(r) \propto r^{-2}$ as quoted in the main text. This behaviour is not permitted asymptotically due to hydrodynamic restrictions.

- We then study the asymptotic critical dynamics for energy dependent matrix elements (146) with $\eta = \infty$, and $0 < \mu < \infty$. We show that in this case $F(r) \sim r^{-1/2}$, behaviour consistent with hydrodynamics.

### C.1  Critical point with energy independent matrix elements

Here we study the equation defined in the main text, specifically

$$\frac{\partial f_\uparrow}{\partial r} = W^2 \frac{\partial^2 f_\uparrow}{\partial e^2} - f_\uparrow q_\uparrow(e,r),$$

$$f_\uparrow\left(e, -\tfrac{1}{2}\right) = \delta\left(e - \tfrac{1}{2}W\right),$$

(154)

for the loss function

$$q_\uparrow(e,r) \sim \frac{1}{\sqrt{4\lambda r}} \exp\left(-\frac{(e + \tfrac{1}{2}W)^2}{2s_e^2(r)}\right),$$

(155)

where $s_e(r) = W\sqrt{2r+1}$.

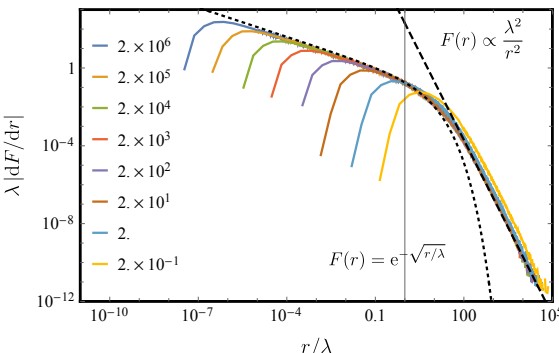

Figure 11: *Decay in $F(r)$ for energy independent matrix elements:* Values of $\lambda|dF/dr|$ are plotted versus $r/\lambda$, these are obtained by numerically solving (154) and (156). The point $r/\lambda = 1$ is marked with a vertical grey line. For $r/\lambda < 1$, the behaviour is consistent with $F(r) = \exp(-\sqrt{r/\lambda})$ (dotted line). For $r/\lambda > 1$, the decay is slower $F(r) \propto (\lambda/r)^2$ (dashed). Different series correspond to different values of $\lambda$ (legend inset).

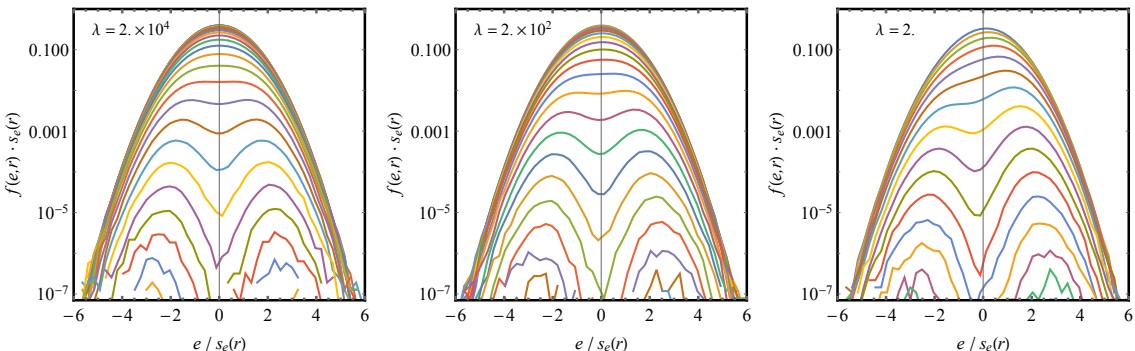

Figure 12: *Decay in $F(r)$ for energy independent matrix elements:* the distributions $f_\sigma(e, r)$ are plotted for log-spaced intervals of $r$, using the same numerical solutions to (154) and (156) as Fig 11. In each case it is clear that at large ranges the distribution is depleted at energies $e \lesssim s_e(r)$.

We numerically solve these equations by stochastic sampling of trajectories. In Fig 11 we plot $dF/dr$ for different values of the parameter $\lambda$ where as before

$$F(r) = \int de f_\uparrow(e, r). \tag{156}$$

We see that for all trajectories the initial decay at small $r \lesssim \lambda$ is consistent with the approximate solution $F(r) = \exp(-\sqrt{r/\lambda})$ (grey vertical line marks $r = \lambda$) at which there is a crossover to $F(r) \propto r^{-2}$ behaviour. For these equations this latter behaviour continues asymptotically.

In Fig. 12 we show the variation of $f_\uparrow(e, r)$ with $e$, specifically we plot $f_\uparrow(e, r)$ for a series of fixed log-spaced values of $r$. For clarity we also re-scale $e$ by the width of the distribution $s_e(r) = W\sqrt{2r + 1}$ (i.e. so that for $\lambda = \infty$ the plots would collapse for all $r$). From these plot it is clear that the centre of the distribution is depleted faster than the mean, that is $f_\uparrow(0, r)$ decays asymptotically faster than $F(r)$. This behaviour is exhibited for $r \gg \lambda$ and violates the approximation scheme of Sec. 3.2.4.

## C.2 Critical point with energy dependent matrix elements

We now study the same loss-diffusion equation (153) for dynamics in the crossover region with energy dependent matrix elements. Specifically we now set

$$q_\uparrow(e,r) \sim \frac{1}{\sqrt{4\lambda r}} \exp\left(-\frac{e^2}{\zeta_c r\mu^2} - \frac{(e+\frac{1}{2}W)^2}{2s_e^2(r)}\right). \tag{157}$$

for some finite $\mu$ in the range $0 < \mu < \infty$.

To simplify the problem we make several approximations which do not alter the asymptotic behaviour of these equations. First, as the width of $q_\sigma$ is asymptotically smaller (in $r$) than $s_e(r)$, for $r \gg \lambda$ we can approximate $q_\uparrow(e,r)$ with a delta function placed at the origin with weight

$$\gamma = \int de\, q_\uparrow(e,r) = \frac{W\mu}{\sqrt{\frac{\lambda}{\zeta_c\pi}(4W^2 + \zeta_c\mu^2)}} + O(r^{-1}). \tag{158}$$

Second, we neglect the sub-leading $r$-dependent correction to $\gamma$, and thirdly we neglect the initial energy offset of $f_\uparrow$. This yields the equation

$$\frac{\partial f_\uparrow}{\partial r} = W^2 \frac{\partial^2 f_\uparrow}{\partial e^2} - \gamma f_\uparrow \delta(e), \tag{159}$$

with boundary condition $f_\uparrow(e, r=0) = \delta(e)$.

To solve this equation we decompose $f_\uparrow$ as

$$f_\uparrow(e,r) = \sum_{n=0}^{\infty} f_n(e,r), \tag{160}$$

which satisfy the equations

$$\frac{\partial f_0}{\partial r} = W^2 \frac{\partial^2 f_0}{\partial e^2}, \tag{161}$$

with boundary condition $f_0(e, r=0) = \delta(e)$ for $n = 0$ and

$$\frac{\partial f_n}{\partial r} = W^2 \frac{\partial^2 f_n}{\partial e^2} - \gamma f_{n-1}\delta(e), \tag{162}$$

with boundary condition $f_n(e, r=0) = 0$ for $n > 0$. With this $f_0$ is straightforwardly identified

$$f_0(e,r) = \frac{e^{-e^2/(4rW^2)}}{\sqrt{4\pi rW^2}}, \tag{163}$$

and it further follows that for $n > 0$

$$f_n(e,r) = -\gamma \int_0^r ds\, f_0(e, r-s) f_{n-1}(0,s) \tag{164}$$

this equation is obtained by simply treating $f_{n-1}(0,s)$ as a source term for $f_n$, in accordance with (162), and integrating with the heat equation Kernel $f_0$. To make progress we note that it is sufficient to obtain the $f_n(0,s)$, which are related by a recursion relation

$$f_n(0,r) = -\gamma \int_0^r ds\, \frac{1}{\sqrt{4W^2\pi(r-s)}} f_{n-1}(0,s), \tag{165}$$

and related to our desired result, $F(r)$, by

$$F(r) = \sum_{n=0}^{\infty} \int \mathrm{d}e f_n(e,r) = 1 - \gamma \sum_{n=1}^{\infty} \int_0^r \mathrm{d}s f_{n-1}(0,r), \tag{166}$$

where we have substituted (164).

Solving this recursion relation (165) yields

$$f_n(0,r) = \frac{(-1)^n}{\gamma \ell \, \Gamma\left(\frac{n+1}{2}\right)} \left(\frac{r}{\ell}\right)^{\frac{n-1}{2}}. \tag{167}$$

where $\ell = 4W^2/\pi\gamma^2$. The function $F(r)$ is then obtained by substituting (167) into (166), performing the integral

$$\gamma \int_0^r \mathrm{d}s f_{n-1}(0,r) = \frac{(-1)^n}{\Gamma\left(\frac{n+3}{2}\right)} \left(\frac{r}{\ell}\right)^{\frac{n+1}{2}} \tag{168}$$

and recognising the resulting summation as a Taylor series, this yields

$$F(r) = \mathrm{e}^{r/\ell} \, \mathrm{Erfc}\left(\sqrt{r/\ell}\right), \tag{169}$$

where

$$\mathrm{Erfc}(x) = 1 - \frac{1}{\sqrt{\pi}} \int_{-x}^x \mathrm{e}^{-t^2} \mathrm{d}t \tag{170}$$

is the usual complementary error function. From (169) it follows that $F(r)$ decays asymptotically as

$$F(r) \sim \sqrt{\frac{\ell}{\pi r}} = \frac{2W}{\pi\gamma\sqrt{r}}, \tag{171}$$

as quoted in the main text. The constant pre-factor here is liable to be altered by the simplifications we made earlier in the calculation, however the asymptotic behaviour $F(r) \propto r^{-1/2}$ is robust.

## D   Linear drift of the deviation from thermal behaviour

In this appendix we derive (116) from the main text

$$W_\delta(L) \approx W_c \mathrm{e}^{-\ell_\delta/(L+1)}, \tag{172}$$

where $\ell_\delta$ is some $\delta$ dependent constant, and $W_\delta(L)$ is defined as the disorder strength at which the time averaged correlator $[\overline{C}_{zz}]$ deviates from thermal behaviour by some small amount $\delta$

$$[\overline{C}_{zz}](W_\delta) = \delta \ll 1. \tag{173}$$

Recalling that $[\overline{C}_{zz}] = F(L/2)$ and using the form (90) for $F(r)$ on the thermal side

$$\delta = \exp\left(-\sqrt{\frac{\pi|\xi(W_\delta)|}{4\lambda(W_\delta)}} \, \mathrm{Erfi}\left(\sqrt{\frac{L}{2|\xi(W_\delta)|}}\right)\right), \tag{174}$$

where we have explicitly labelled disorder dependence of the correlation length $\xi$ and the resonance length $\lambda$. We use

$$\xi(W) \approx \frac{1}{\log(W/W_c)}, \tag{175}$$

whereas $\lambda$ is given by (82).

Let us extract from (174) how $W_\delta$ varies with $L$. Away from the crossover region the imaginary error function can be written in terms of more familiar functions

$$\text{Erfi}(\sqrt{x}) = \frac{e^x}{\sqrt{\pi x}}\left(1 + O(x^{-1})\right).$$
(176)

Substituting both (176) and $\lambda(W_\delta) = (W_\delta/W_c)^2 \lambda(W_c)$ into (174) and rearranging we obtain

$$\frac{L}{|\xi(W_\delta)|} + \log\frac{W_\delta}{W_c} = 2\log\left(\frac{\sqrt{2L\lambda(W_c)}}{|\xi(W_\delta)|}|\log\delta|\right) + O\left(\frac{2|\xi(W_\delta)|}{L}\right).$$
(177)

Consider the RHS of (177): for sufficiently small $\delta$ we are far from the crossover $L \gg |\xi|$ and the corrections may be neglected. Now consider the leading term on the RHS of (177): this term exhibits weak logarithmic dependence of $L$, and, recalling that $\xi(W_\delta) \approx 1/\log(W_\delta/W_c)$, doubly logarithmic dependence on $W_\delta$, thus to first approximation the RHS may be replaced by a (negative) constant $-\ell_\delta$:

$$\frac{L}{|\xi(W_\delta)|} + \log\frac{W_\delta}{W_c} = -\ell_\delta.$$
(178)

Then, again using $\xi(W_\delta) \approx 1/\log(W_\delta/W_c)$, by rearranging we obtain the desired result (172).

This function is approximately linear for sufficiently small $L$. To see this, note that the RHS of (116) has an inflection point at $L = \ell_\delta/2 - 1$, and thus has zero curvature at this point. Taylor expanding about the inflection point and demanding that the cubic term is not larger than the linear term reveals the approximate linearity to persist for $L + 1 \lesssim \ell_\delta(1/2 + \sqrt{3/4})$.

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
