# Peer review of "A constructive theory of the numerically accessible many-body localized to thermal crossover"

_SciPost Physics, doi:SciPost Phys. 12, 201 (2022)_

## Round 1 · Referee Report · Anonymous (Referee 1) · 2022-1-19

Report

In this work, the authors introduce a model for the description of the finite-size crossover properties in the vicinity of the putative MBL transition in one spatial dimension. Recent works in the field of MBL, in particular concerning exact diagonalization numerics for small system sizes, have raised serious questions on the MBL transition or even on the fundamental aspect of the very existence of the MBL phase. The key open problem at this point has been so far to understand the finite-size properties of strongly disordered systems when getting closer towards the (finite-size) MBL transition. In this context, the present work represents an important contribution to the field, especially because it is of analytical nature, and is very timely.

The manuscript is very well written and presents a variety of results, which all appear sound. Importantly, the results cover a very broad range of parameter regimes as well as physical observables. Further, the authors show that their analytical predictions match many of the numerical finite-size observations in the vicinity of the MBL transition, which is remarkable in my opinion. Consequently, this extensive manuscript certainly deserves publishing in SciPost. I just have a few minor comments, which might help to further improve the manuscript

  1. To my taste the results of Refs. [1,2] are sometimes referred to in a bit subjective way, which doesn't always appear appropriate. For instance, on page 2 it is written that both Refs. [1,2] "claim that the numerical data precludes the possibility of an MBL phase altogether". I don't think that both papers actually claim that, the claims are partly weaker. Maybe it would be possible to weaken such statements slightly. By the way, both are now also published in journals.
  2. When I understand correctly (see for instance Eq. (43)), the local perturbation studied in this manuscript is always in a weak coupling regime. Is that correct? In any case it would be good to clarify that prominently.
  3. Section 2.3 on the thermal phase is rather brief. Although (as the authors say) the RM is not applicable, in the next sentence they say that RM still holds. This sounds a bit confusing and it would be good to clarify this. Further, the authors mention "almost-l-bits" without any reference. I feel that it is not clear what kind of l-bits these should be.

As soon as these minor aspects are taken into account, I would certainly recommend publication in SciPost.

---

## Round 1 · Referee Report · Anonymous (Referee 2) · 2022-2-1

Strengths

  • This is an extremely important contribution to the literature on finite-size effects in many-body localization.
  • The paper is clearly written and the study is highly innovative.

Weaknesses

  • I am not aware of any major weaknesses, I think the paper should be published as is.

Report

I think this is a very important paper and should be published as is.

---

## Round 2 · Author Response

We thank both referees for their time and careful reading of our manuscript. Report 2 recommends publication as is. We detail the changes we have made to address the comments raised in report 1 below.

We hope with this response, we have satisfied the queries of the referees and editors.

  1. “To my taste the results of Refs. [1,2] are sometimes referred to in a bit subjective way, which doesn't always appear appropriate. For instance, on page 2 it is written that both Refs. [1,2] "claim that the numerical data precludes the possibility of an MBL phase altogether". I don't think that both papers actually claim that, the claims are partly weaker. Maybe it would be possible to weaken such statements slightly. By the way, both are now also published in journals.”

We have substituted the line “claim that the numerical data precludes the possibility of an MBL phase altogether” to “argue that finite size numerics is inconsistent with the existence of MBL in the thermodynamic limit”, as paraphrased from the conclusion of Ref [2].

  1. “When I understand correctly (see for instance Eq. (43)), the local perturbation studied in this manuscript is always in a weak coupling regime. Is that correct? In any case it would be good to clarify that prominently.”

If by “weak coupling” the referee means \Omega, W >> J, yes, we are always working in this regime. This is where both the MBL finite size crossover and, at even stronger disorder, it is believed the MBL transition also occurs (they certainly do not occur outside of this regime). This regime is introduced in the first sentence “Interacting one-dimensional quantum systems generically many-body localise (MBL) in the presence of strong disorder.”. It is further restated below Eq.4 in which the model is introduced “We assume two key properties of H(t): (i) it has no global conservation laws, and (ii) for some finite \Omega, W >> J, the model is Floquet many-body localised, as per Ref. [71]. The specific form of H(t) is otherwise unimportant.”

If the referee instead means perturbatively weak coupling, then no, we do not make this assumption. Our results are non-perturbative in the probe spin coupling, this is important in allowing us to capture the effect of the non-perturbatively corrected states or “resonances”. We note this picture of resonance formation we assume is validated in a more controlled model in Ref [72].

  1. “Section 2.3 on the thermal phase is rather brief. Although (as the authors say) the RM is not applicable, in the next sentence they say that RM still holds. This sounds a bit confusing and it would be good to clarify this. Further, the authors mention "almost-l-bits" without any reference. I feel that it is not clear what kind of l-bits these should be.”

Precisely which of the predictions of the RM we expect to hold, and for what regime of time is clarified by the sentence “Despite being generally inapplicable, the early time predictions of the RM are found to hold even in the thermal regime” and following equations. The almost l-bits are defined in the manuscript Eq. 60 and surrounding text. Where we write “almost-lbits [are] operators [which] have the same properties as l-bits (mutually commuting exponentially localised etc.), but only “almost commute” with the Hamiltonian: |[H,\tau]|<\omega_\xi.

We are not the first to propose such objects. We have included a citation to the earlier work Ref. [73] in which similar objects were proposed.

---

## Round 2 · List of Changes

• Substituted the line “claim that the numerical data precludes the possibility of an MBL phase altogether” to “argue that finite size numerics is inconsistent with the existence of MBL in the thermodynamic limit”.

  • Included citation to Ref. [73]

---

## Editorial Decision

published